# Mendelian randomization suggests a bidirectional, causal relationship between physical inactivity and adiposity

**Germán D Carrasquilla[1]\*, Mario García-Ureña[1], Tove Fall[2], Thorkild IA Sørensen[1,3], Tuomas O Kilpeläinen[1]**

[1]Novo Nordisk Foundation Center for Basic Metabolic Research, Faculty of Health and Medical Sciences, University of Copenhagen, Copenhagen, Denmark; [2]Molecular Epidemiology, Department of Medical Sciences, and Science for Life Laboratory, Uppsala University, Uppsala, Sweden; [3]Department of Public Health, Section of Epidemiology, Faculty of Health and Medical Sciences, University of Copenhagen, Copenhagen, Denmark

**\*For correspondence:**
german.carrasquilla@sund.ku.dk

**Competing interest:** The authors declare that no competing interests exist.

**Abstract** Physical inactivity and increased sedentary time are associated with excess weight gain in observational studies. However, some longitudinal studies indicate reverse causality where weight gain leads to physical inactivity and increased sedentary time. As observational studies suffer from reverse causality, it is challenging to assess the true causal directions. Here, we assess the bidirectional causality between physical inactivity, sedentary time, and adiposity by bidirectional Mendelian randomization analysis. We used results from genome-wide association studies for accelerometer-based physical activity and sedentary time in 91,105 individuals and for body mass index (BMI) in 806,834 individuals. We implemented Mendelian randomization using CAUSE method that accounts for pleiotropy and sample overlap using full genome-wide data. We also applied inverse variance-weighted, MR-Egger, weighted median, and weighted mode methods using genome-wide significant variants only. We found evidence of bidirectional causality between sedentary time and BMI: longer sedentary time was causal for higher BMI [beta (95% CI) from CAUSE method: 0.11 (0.02, 0.2), p = 0.02], and higher BMI was causal for longer sedentary time (0.13 (0.08, 0.17), p = 6.3 x 10[-4]). Our analyses suggest that higher moderate and vigorous physical activity are causal for lower BMI (moderate: –0.18 (-0.3,–0.05), p = 0.006; vigorous: –0.16 (-0.24,–0.08), p = 3.8 × 10[-4]), but indicate that the association between higher BMI and lower levels of physical activity is due to horizontal pleiotropy. The bidirectional, causal relationship between sedentary time and BMI suggests that decreasing sedentary time is beneficial for weight management, but also that targeting adiposity may lead to additional health benefits by reducing sedentary time.

## Editor's evaluation

This paper is of potential interest to those researchers and clinicians working in the area of physical activity and obesity. The authors have presented strong evidence of a causal relationship between physical activity and higher BMI. The conclusions that have been made are supported by the data and are translationally relevant.

## Introduction

Obesity and physical inactivity are major risk factors for a number of chronic diseases, such as type 2 diabetes, cardiovascular diseases and several types of cancer. Today's epidemic of obesity and

sedentary lifestyle is thus a major burden on public health systems worldwide (*GBD 2019 Risk Factors Collaborators, 2020*).

Many observational studies suggest that physical inactivity and increased sedentary time are associated with a higher risk of obesity (*Lee et al., 2010*; *Du et al., 2013*; *Silva et al., 2019*; *Myers et al., 2017*). However, other studies have indicated a reverse effect, where obesity leads to physical inactivity and increased sedentary time (*Petersen et al., 2004*; *Mortensen et al., 2006*; *Bak et al., 2004*; *Barone Gibbs et al., 2020*; *Ekelund et al., 2008*; *Myers et al., 2017*). Furthermore, randomized clinical trials of physical activity interventions have indicated that the causal effects of physical activity on body weight are modest (*Church et al., 2009*; *Rosenkilde et al., 2012*; *Golubic et al., 2015*) compared to the strong inverse relationship between physical activity and body weight observed in cross-sectional epidemiological studies. This suggests that the observational results may be affected by bias, such as reverse causality or confounding by other lifestyle or environmental factors (*Schnurr et al., 2021*). To date, the causal relationships between physical inactivity, sedentary time and adiposity remain unclear and warrant further investigation. It is important to assess the causal relationship with adiposity separately for physical activity and sedentary time, as individuals can be physically active even if they engaged a substantial part of their time to sedentary behaviors, and vice versa (*Panahi and Tremblay, 2018*).

Mendelian randomization is a powerful method to minimize the influence of reverse causality and confounding on causal estimates derived from observational data. Since genotypes are randomly allocated at conception, genetic alleles associated with physical activity, sedentary behavior, and body mass index (BMI) can be used to assign individuals according to higher or lower mean levels of these exposures in a randomized manner (*Figure 1*).

Here, we aimed to assess the causality between the genetic liability of physical inactivity, sedentary time and BMI by applying bidirectional Mendelian randomization analyses on summary results of accelerometer-based physical activity and sedentary time for 91,105 adults and of BMI for 806,834 adults.

## Results

We used the Mendelian randomization CAUSE method to take advantage of the full genome-wide summary results (*Morrison et al., 2020*). We found evidence of causality between higher vigorous and moderate physical activity and lower BMI (p = $3.8 \times 10^{-4}$ and p = 0.006, respectively), and between more sedentary time and higher BMI (p = 0.02) (*Table 1*, *Figure 2*, and *Appendix 1—table 1* and *Appendix 1—table 2*). In absolute units, we approximate that each one hour daily increase in moderate physical activity or decrease in sedentary time causally decreased BMI by 0.27 kg/m² (~0.8 kg) or 0.14 kg/m² (~0.4 kg) (*Appendix 1—table 3*). The median shared effect, which reflects the effect induced by correlated horizontal pleiotropy, ranged from –0.01 to 0 for all trait pairs, indicating that there was no bias induced by horizontal pleiotropy. The low q values under the causal model (q = 0.18–0.20), which reflect the proportion of variants that show correlated horizontal pleiotropy, also suggested that horizontal pleiotropy was limited. We checked for the existence of possible outlier variants that could have a large effect on the exposure but no effect on the outcome, by producing a scatter plot for CAUSE test statistics on the causal association between BMI and sedentary time (*Appendix 1—figure 1*). While CAUSE is not sensitive to outliers as such, the existence of outlier variants could provide evidence against the causal model. However, we found no evidence of outlier variants. As we found high shared model q values in our CAUSE analyses (q=0.7-0.9), which may suggest that the genetic effects between the two traits are highly correlated and causality will be difficult to establish, we also assessed the genetic correlation between BMI and sedentary time. However, the genetic correlation analysis suggested only a modest genetic correlation between BMI and sedentary time ($r_g$=0.27), which provides further reassurance that our CAUSE results are valid.

In the reverse direction, we found no evidence of a causal effect of BMI on vigorous physical activity (p = 0.35) or moderate physical activity (p = 0.31) using CAUSE (*Table 1*, *Figure 2*, and *Appendix 1—table 1* and *Appendix 1—table 2*). However, we found evidence of a causal effect of BMI on more sedentary time (p = $6.3 \times 10^{-4}$), indicating bidirectional causality between the traits. The median shared effect in the causal association between BMI and sedentary time was zero and the q value was 0.18, suggesting that the causal association between sedentary time and BMI was unlikely to be

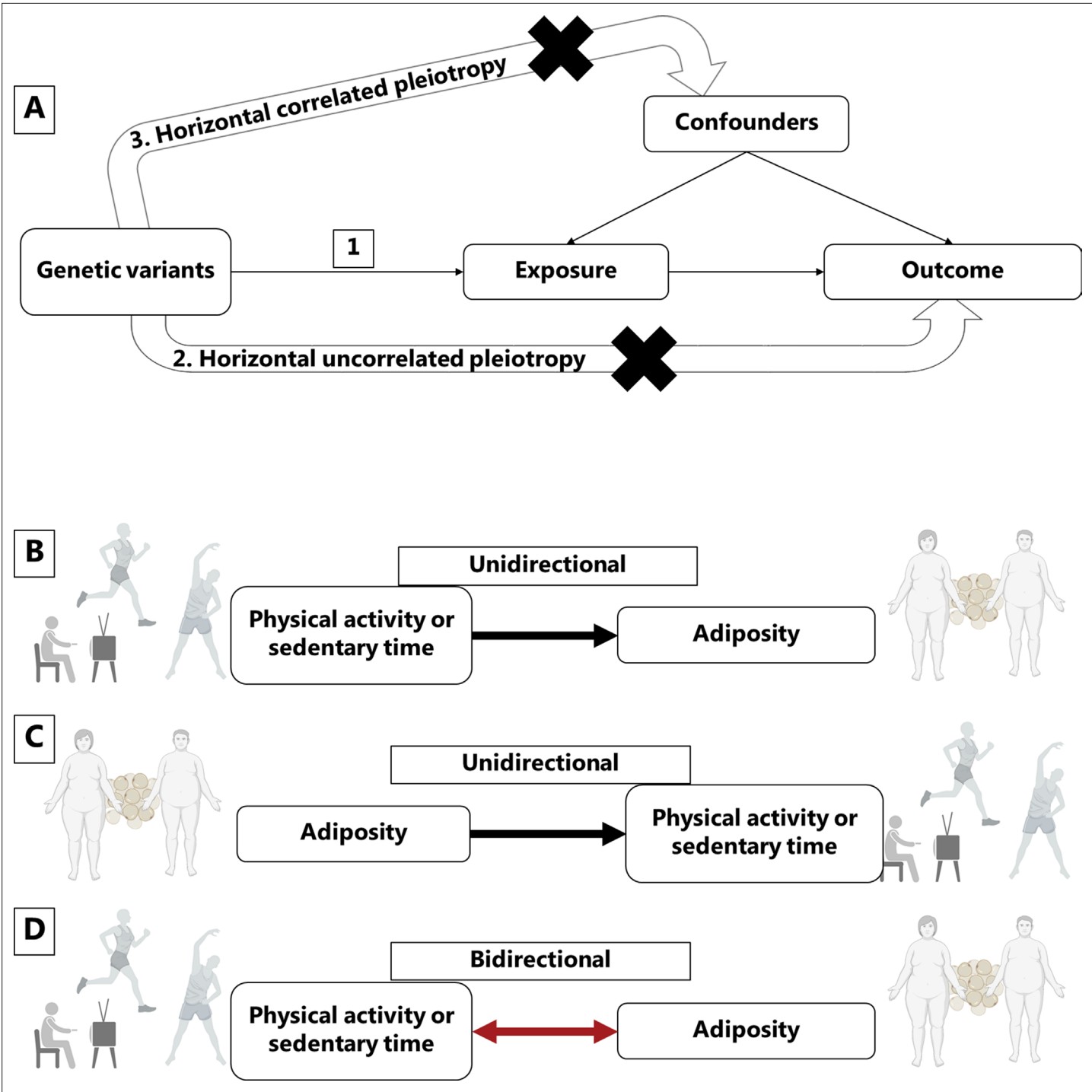

**Figure 1.** Mendelian randomization assumptions and directional associations between physical activity, sedentary time and adiposity. Panel A shows Mendelian randomization assumptions when estimating the causal association between a given exposure and outcome: (1) The instruments are associated with the exposure, (2) the instruments do not cause the outcome independently from the exposure (horizontal uncorrelated pleiotropy), and (3) the effects of the exposure on the outcome are not influenced by any confounders (horizontal correlated pleiotropy). Panel B indicates a unidirectional causal effect of physical activity or sedentary time on adiposity, whereas panel C indicates a unidirectional causal effect of adiposity on physical activity or sedentary time. Panel D indicates a bidirectional causal relationship where physical activity or sedentary time has a causal effect on adiposity, but at the same time, adiposity has a causal effect on physical activity or sedentary time. Figure icons were created with BioRender.com.

**Table 1.** Results for Mendelian randomization analyses using the CAUSE method.

**Causal model better fit for the data**

| Direction | Median causal effect (95% CI) | Median q (CI) | *P* causal vs sharing |
|---|---|---|---|
| Vigorous PA→ BMI | –0.16 (-0.24,–0.08) | 0.19 (0, 0.86) | $3.8 \times 10^{-04}$ |
| Moderate PA→ BMI | –0.18 (-0.3,–0.05) | 0.20 (0.01, 0.86) | 0.006 |
| Sedentary time → BMI | 0.11 (0.02, 0.20) | 0.19 (0, 0.86) | 0.02 |
| BMI → Sedentary time | 0.13 (0.08, 0.17) | 0.18 (0, 0.85) | $6.3 \times 10^{-4}$ |

**Sharing model better fit for the data**

| Direction | Median shared effect (CI) | Median q (CI) | *P* causal vs sharing |
|---|---|---|---|
| BMI → Vigorous PA | –0.16 (-0.19,–0.14) | 0.90 (0.77, 0.98) | 0.35 |
| BMI → Moderate PA | –0.14 (-0.19,–0.11) | 0.77 (0.55, 0.95) | 0.31 |

BMI, Body mass index (BMI); SE, standard error; N SNPs, number of single-nucleotide polymorphism; P, P-value; PA, physical activity; IVW, inverse variance weighted; CI, 95% confidence interval.

The results display the data according to the goodness-of-fit for the causal or the sharing model. The median q value indicates the proportion of variants with correlated pleiotropy.

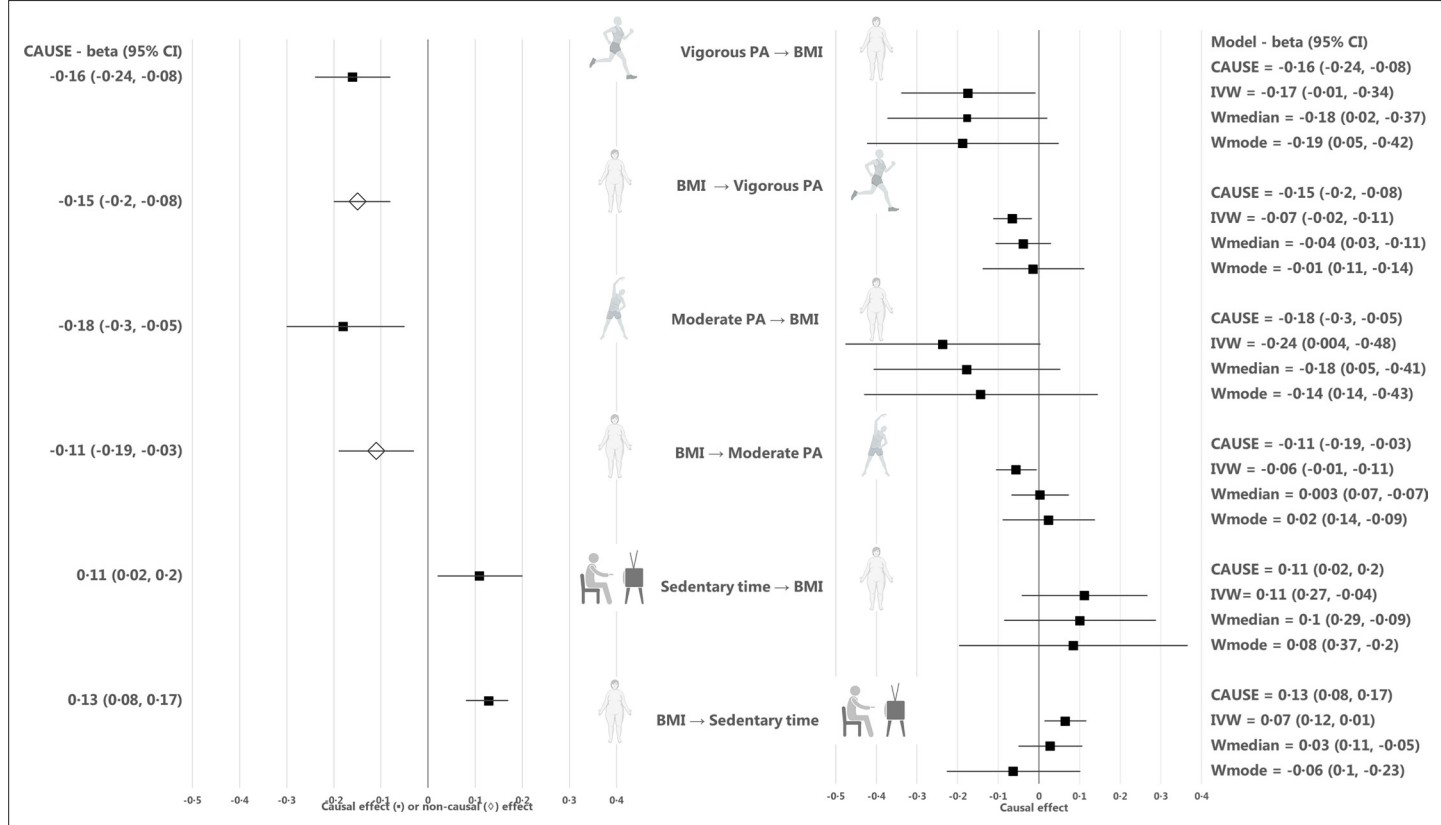

**Figure 2.** Causal estimates for Mendelian randomization analyses using the CAUSE, inverse-variance-weighted (IVW), weighted median, and weighted mode methods. Median causal estimates for Mendelian randomization analyses using the CAUSE method are shown in the left panel and mean causal estimates from the inverse variance weighted (IVW), weighted median (Wmedian) and weighted mode (Wmode) methods are shown in the right panel after outlier removal and accounting for horizontal pleiotropy. A diamond (◊) in the estimate for CAUSE indicates that the sharing model fit the data better than the causal model, that is that the association between the traits was more likely to be explained by horizontal correlated pleiotropy than causality. PA, physical activity; BMI, body mass index. Figure icons were created with BioRender.com.

**Table 2.** Mendelian randomization results for inverse variance weighted, weighted median, weighted mode, and MR-Egger methods.

| Direction | Vigorous physical activity → BMI | | | | Moderate physical activity → BMI | | | | Sedentary time → BMI | | | |
|---|---|---|---|---|---|---|---|---|---|---|---|---|
| MR method | beta | SE | p-value | N SNPs | beta | SE | p-value | N SNPs | beta | SE | p-value | N SNPs |
| IVW | −0.17 | 0.08 | 0.04 | 5 | −0.24 | 0.12 | 0.05 | 3 | 0.11 | 0.08 | 0.16 | 5 |
| Weighted median | −0.18 | 0.10 | 0.08 | 5 | −0.18 | 0.12 | 0.13 | 3 | 0.10 | 0.10 | 0.29 | 5 |
| Weighted mode | −0.19 | 0.12 | 0.19 | 5 | −0.14 | 0.15 | 0.43 | 3 | 0.08 | 0.14 | 0.59 | 5 |
| MR-Egger | 1.33 | 2.21 | 0.59 | 5 | 0.12 | 0.45 | 0.84 | 3 | 0.49 | 0.36 | 0.26 | 5 |
| Direction | BMI → Vigorous physical activity | | | | BMI → Moderate physical activity | | | | BMI → Sedentary time | | | |
| IVW | −0.07 | 0.02 | 0.01 | 57 | −0.06 | 0.03 | 0.03 | 55 | 0.07 | 0.03 | 0.01 | 57 |
| Weighted mode | −0.01 | 0.06 | 0.83 | 57 | 0.02 | 0.06 | 0.68 | 55 | −0.06 | 0.08 | 0.46 | 57 |
| Weighted median | −0.04 | 0.03 | 0.27 | 57 | 0.003 | 0.04 | 0.93 | 55 | 0.03 | 0.04 | 0.48 | 57 |
| MR-Egger | 0.13 | 0.07 | 0.06 | 57 | 0.16 | 0.07 | 0.04 | 55 | −0.02 | 0.08 | 0.76 | 57 |

BMI, Body mass index (BMI); SE, standard error; N SNPs, number of single nucleotide polymorphism; IVW, inverse variance weighted; CI, 95% confidence interval.

The results display the data according to the goodness-of-fit for the causal or the sharing model. The median q value indicates the proportion of variants with correlated pleiotropy. CI, confidence interval; PA, physical activity; BMI, body mass index; P, P-value.

biased by horizontal pleiotropy. In absolute units, we approximate that each $kg/m^2$ (~3 kg) increase in BMI was causally associated with a 3.5 min increase in sedentary time per day (*Appendix 1—table 3*).

We also estimated the causal effects of moderate physical activity, vigorous physical activity and sedentary time on BMI with four commonly used Mendelian randomization methods, including IVW, Egger, weighted median and weighted mode methods. Due to the low number of independent, genome-wide significant loci for vigorous physical activity, moderate physical activity and sedentary time that were present in the GWAS results for BMI, we used a less stringent threshold of $p < 5 \times 10^{-7}$ to identify genetic instruments for these traits, resulting in 5, 3, and 5 independent loci, respectively. The directions of causal estimates were consistent with the findings from CAUSE, but the evidence for causality was weaker (*Table 2*, *Figure 2*, Appendix 1, and *Appendix 1—table 4*). To estimate the causal effect of BMI on moderate physical activity, vigorous physical activity and sedentary time, we used genome-wide significant BMI loci ($p < 5 \times 10^{-8}$) as instruments (n = 57, n = 55 and n = 57, respectively). Again, the directions of causal estimates were consistent with the CAUSE results, but the associations were weaker (*Table 2*, *Figure 2*, Appendix 1, and *Appendix 1—table 5*).

The results from CAUSE analyses for BFP were consistent with those for BMI, that is there was evidence of a causal effect of higher vigorous and moderate physical activity on lower BFP ($p = 1.4 \times 10^{-6}$ and $p = 0.004$, respectively), and a causal effect of more sedentary time on higher BFP ($p = 0.009$) (*Figure 2*, *Appendix 1—table 6* and *Appendix 1—table 7*, *Appendix 1—figure 2*). We found no evidence of a causal relationship between physical activity or sedentary time and measures of central adiposity, including $WC_{adjBMI}$ and $WHR_{adjBMI}$ (*Figure 2*, *Appendix 1—table 6* and *Appendix 1—table 7*, *Appendix 1—figure 2*).

## Discussion

The present Mendelian randomization analyses suggest a bidirectional causal relationship between higher sedentary time and higher BMI, implying that decreasing sedentary time is beneficial for weight management, but also that reducing adiposity may lead to additional health benefits by reducing sedentary time. The analyses also suggest there is a causal association between higher levels of physical activity and lower BMI, supporting the view that preventive programs for increasing physical activity and decreasing sedentary time are beneficial for weight management.

Based on the causal effect size in our analysis, we estimated that each 1 hr daily increase in moderate physical activity or 1 hr decrease in sedentary time was associated with a 0.27 $kg/m^2$ (~ 0.8 kg) or 0.14 $kg/m^2$ (~0.4 kg) decrease in BMI or body weight, respectively. Our results also suggest that each 1 $kg/m^2$ (~3 kg) higher BMI increases daily sedentary time by ~3.5 min, but do not suggest a causal effect of BMI on physical activity (Appendix 1 and *Appendix 1—table 3*). Our results are

well in accordance with a previous observational study that aimed to assess the bidirectional relationship between physical activity and weight change during a 10-year period (*Barone Gibbs et al., 2020*). Examining associations between accelerometer-based activity measures and weight change in 866 men and women, the study suggested a bidirectional relationship where higher sedentary time at baseline increased 10-year weight gain and higher baseline weight was associated with an unfavorable 10-year change in sedentary time. The effect sizes indicated that 1 hr lower sedentary time at baseline was associated with ~0.3 kg decrease in body weight over the 10-year follow up, whereas 3 kg higher baseline body weight was associated with ~2 min increase in sedentary time, closely resembling the causal estimates we observed in the present Mendelian randomization analyses. Our results are also in accordance with randomized clinical trials of physical activity interventions which generally suggest that increasing physical activity leads to a moderate loss of body weight in overweight or obese participants (*Church et al., 2009*; *Rosenkilde et al., 2012*; *Golubic et al., 2015*; *Kim et al., 2019*; *Biddle et al., 2015*). Resembling the causal estimates we observed, meta-analyses of randomized clinical trials have suggested weight losses ranging from 0.3 kg to 1.8 kg for various physical activity interventions ranging from 2 to 52 weeks (*Twells et al., 2021*). However, it is important to note that the causal estimates from Mendelian randomization are not fully comparable to those from randomized clinical trials, because they represent lifelong effects rather than effects lasting a defined length of an intervention, and furthermore, physical activity interventions may operate on body weight through other pathways than those affected by the genotypes. The causal effect of higher adiposity on sedentary time has not been to date assessed in randomized clinical trials, likely due to the ethical and practical limitations of performing such a study.

In a previous Mendelian randomization analysis of adult populations, evidence for a causal, bidirectional relationship between overall activity levels and higher BMI was observed using the maximum likelihood method, but the results showed evidence of horizontal pleiotropy that could not be fully accounted for and the role of activity intensity level remained unclear (*Doherty et al., 2018*). Here, using a method that takes advantage of full genome-wide summary results and corrects for sample overlap between the exposure and the outcome traits to maximize statistical power and correct for pleiotropy, we showed that the causal bidirectional relationship is particularly evident for the relationship between sedentary time and adiposity. Our results may also be compared with two independent one-sample Mendelian randomization studies performed in children (*Richmond et al., 2014*; *Schnurr et al., 2018*). The first study, including 4296 children at 11 years of age from the United Kingdom, indicated a causal association between higher BMI and lower accelerometer-based moderate and moderate-to-vigorous physical activity and more sedentary time (*Richmond et al., 2014*). The second study, including 679 children at age 3–8 years from Denmark, also indicated that higher BMI is causal for higher accelerometer-based sedentary time, but did not find a causality for moderate or moderate-to-vigorous physical activity (*Schnurr et al., 2018*). Consistent with the latter study of children, our results indicate a causal effect of BMI on sedentary behavior, but not on physical activity, in adults. The differences between studies could be due to different applied methods, or methodological limitations, such as weak instrument bias when smaller sample sizes are used, which may lead to estimated causal effects towards the observational association. One could also expect differences between children and adults given the distinct patterns by which they engage in physical activity. For example, while physical activity in adults consists of commuting, occupational and structured leisure-time activities, children primarily engage in spontaneous, play-oriented activities. Higher BMI leads to higher perceived exertion during physical activity (*Groslambert and Mahon, 2006*), which could reduce the natural inclination of children to engage in play-oriented activities, whereas adults exert more conscious control over their daily activities.

The strengths of the present studies include the use of genome-wide summary results for objectively measured physical activity and sedentary time, which avoided misreporting bias evident for self-reported measures, as well as the use of newly developed Mendelian randomization method that utilizes full genome-wide summary results to improve statistical power, correct for sample overlap, and assess horizontal pleiotropy, successfully applied in recent Mendelian randomization studies (*Jäger et al., 2020*; *Mitchell et al., 2020*). The limitations are that we cannot exclude other sources of bias in the measurement of physical activity and sedentary time that could influence the observed causal estimates, including the observer effect and the limited 7-day period of the measurement, which may not be representative of long-term activity habits. Furthermore, even if we used the largest available

data on objectively measured physical activity, the statistical power was limited, as very few genome-wide significant loci have thus far been identified. When larger sample sizes for accelerometer-based physical activity become available, the results should be replicated. The present study is also limited in the fact that the findings are not generalizable across different age-groups or populations. Moreover, further research is needed to investigate causal relationships at various BMI thresholds.

In conclusion, the present Mendelian randomization analyses indicate a bidirectional causal relationship between higher sedentary time and higher BMI. Thus, decreasing sedentary time is likely to be beneficial for weight management, but reducing adiposity may also lead to additional health benefits by reducing sedentary time. Our analyses also suggest that there is a causal association between higher levels of physical activity and lower BMI, supporting the view that lifelong preventive programs for increasing physical activity and decreasing sedentary time are beneficial for weight management.

## Materials and methods

### Data sources and populations

We used summary results from the largest published genome-wide association studies (GWAS) of objectively assessed physical activity, sedentary behavior, and BMI in individuals of European ancestry. The physical activity GWAS included up to 91,105 individuals for accelerometer-based vigorous physical activity, moderate physical activity, or sedentary time from the UK Biobank (*Klimentidis et al., 2018*; *Doherty et al., 2018*). In these studies, accelerometer was worn continuously for at least 72 hr and up to 7 days. Vigorous physical activity was defined as the fraction of accelerations > 425 milligravities, and moderate physical activity was predicted using a machine-learning method for moderate intensity activity time (*Doherty et al., 2018*). Sedentary time was defined as the time spent in activities with metabolic equivalent of task (MET) ≤1.5 during sitting, lying, or in reclining posture, except for driving and certain non-desk work instances where MET ≤2.5 was applied (*Doherty et al., 2018*). For BMI, we utilized GWAS results from a meta-analysis of the Genetic Investigation of Anthropometric Traits (GIANT Consortium) and the UK Biobank data, including altogether 806,834 individuals of European ancestry (*Pulit et al., 2019*). For Mendelian randomization analyses using the inverse variance-weighted (IVW), weighted median, weighted mode, and MR-Egger regression methods, we used only the GIANT Consortium BMI meta-analysis data of 339,224 individuals without the UK Biobank data to avoid sample overlap between the exposure and outcome traits as these methods are sensitive to bias from overlapping samples (*Locke et al., 2015*).

In addition to BMI, we assessed causal effects of physical activity and sedentary time on body fat percentage (BFP) and on two measures of central adiposity, including waist circumference and waist-hip ratio adjusted for BMI ($WC_{adjBMI}$ and $WHR_{adjBMI}$, respectively). GWAS results for BFP were available from an analysis of 454,633 participants of European ancestry in the UK Biobank (*Elsworth et al., 2020*). GWAS results for $WC_{adjBMI}$ and $WHR_{adjBMI}$ were available from GIANT Consortium meta-analyses of 231,355 and 694,649 individuals of European ancestry, respectively (*Justice et al., 2017*; *Pulit et al., 2019*).

### Mendelian randomization using full genome-wide summary results for the exposure trait

Only few genetic loci have been found to be associated with accelerometer-based moderate physical activity (n = 2), vigorous physical activity (n = 1) or sedentary time (n = 4) at genome-wide significance ($p < 5 \times 10^{-8}$) (*Klimentidis et al., 2018*; *Doherty et al., 2018*), and the loci thus provide a limited power to study causal associations with BMI using Mendelian randomization. The recently published Causal Analysis Using Summary Effect Estimates (CAUSE) Mendelian randomization method (*Morrison et al., 2020*) improves statistical power in such cases, by utilizing full genome-wide summary results instead of genome-wide significant loci only. Furthermore, the CAUSE method is able to correct for sample overlap between the exposure and the outcome trait, which allows using the largest sample sizes available for both traits. CAUSE has also been found to be less prone to identify false positive associations compared to other commonly used Mendelian randomization methods (*Burgess et al., 2019*; *Morrison et al., 2020*).

The CAUSE method calculates the posterior probabilities of the causal effect and the shared (non-causal) effect, where the causal effect reflects the effect of the variants on the outcome trait through

the exposure and the shared effect reflects correlated horizontal pleiotropy (*Figure 1*), that is the effect of the variants on the outcome through confounders. The distinction between a causal effect and correlated horizontal pleiotropy follows the assumption that a causal effect leads to non-zero genetic correlation between the exposure and the outcome where the correlation is driven by all variants associated with the exposure. If only a subset of variants contributes to the genetic correlation between the exposure and the outcome, it is considered the result of correlated horizontal pleiotropy. The CAUSE method also provides an estimate of the proportion of variants that are likely to show correlated horizontal pleiotropy, the q value.

We used the CAUSE settings and procedures originally recommended by the authors (*Morrison et al., 2020*), with the exception of q priors that were set to fit the strictest model possible (q_alpha = 1 and q_beta = 2) in order to avoid false positive findings. A thorough explanation of the steps used to perform CAUSE analysis is included in the supplementary text (Appendix 1).

## Mendelian randomization using genome-wide significant loci for the exposure trait

In addition to the CAUSE method that implements Mendelian randomization analyses using full genome-wide summary results for the exposure trait, we implemented four commonly used Mendelian randomization methods that utilize genome-wide significant loci only: the IVW, MR-Egger, weighted median and weighted mode methods (Appendix 1). We performed sensitivity analyses using Steiger filtering to remove variants that showed stronger association with the outcome than the exposure trait and that were thus not considered suitable as instruments for the exposure trait. To create the genetic instrument for the exposure trait, we only included the lead variants that showed genome-wide significant associations with the trait ($p < 5 \times 10^{-8}$) and with a pairwise linkage disequilibrium (LD) $r^2 < 0.001$ with their neighboring variants, in a window of 10,000 kb. Variants that were not available in the outcome trait GWAS were substituted by their LD proxies ($r^2 > 0.8$). Palindromic variants (A/T, G/C) were excluded. If less than three genetic variants were identified with these parameters, we used a less stringent p-value threshold of $p < 5 \times 10^{-7}$ to identify enough genetic instruments. In order to assess the strength of the genetic instrument, we obtained *F*-statistics for each trait. The analyses were performed using the TwoSampleMR package in R and are described in detail in the Appendix 1 (*Hemani et al., 2018*).

We estimated heterogeneity across the causal estimates of the SNPs using the Meta R package (*Schwarzer et al., 2015*). The causal estimates were considered heterogeneous if the p value for Cochran's Q test was significantly different from zero ($p < 0.05$) and $I^2$ was above 0.25. We assessed bias introduced by horizontal pleiotropy by implementing the Egger's intercept test using the TwoSampleMR package in R (*Hemani et al., 2018*). An Egger's intercept that deviated significantly from zero ($p < 0.05$) was considered as evidence of horizontal pleiotropy. We used the Rucker framework (*Bowden et al., 2018*) to assess whether Egger regression that accounts for horizontal pleiotropy but limits statistical power should be applied instead of the standard IVW model. To visually assess heterogeneity and horizontal pleiotropy, we observed forest plots and funnel plots (*Figure 2*, *Appendix 1—figure 3*). To detect individual pleiotropic variants that might bias the results, we applied the RadialMR package in R using an iterative Cochran's Q method and setting a strict outlier P value threshold of <0.05 (*Bowden et al., 2018*). The iterative Cochran's Q, either IVW's Q or Egger's Q was chosen depending on Rucker framework results. After removing outlier variants detected with RadialMR, we re-run the Mendelian randomization and sensitivity tests and plots, to make sure that the variants introducing horizontal pleiotropy (*Figure 1*) had been removed. The analysis plan for this study is described in the supplementary text (Appendix 2).

The CAUSE method's median posterior probability of the causal effect cannot be easily transformed to absolute units. To convert the causal estimates to absolute units, we calculated a causal effect with weighted median method using independent variants identified in CAUSE that were not removed by the outlier extraction protocol described above, in order to mimic CAUSE control for correlated and uncorrelated pleiotropy (Appendix 1).

## Acknowledgements

This project has received funding from the European Union's Horizon 2020 research and innovation programme under the Marie Sklodowska-Curie grant agreement No 846,502. Novo Nordisk Foundation Center for Basic Metabolic Research is an independent research center at the University of Copenhagen partially funded by an unrestricted donation from the Novo Nordisk Foundation (NNF18CC0034900). Germán D Carrasquilla was supported by a grant from the Danish Diabetes Academy that is funded by the Novo Nordisk Foundation (NNF17SA0031406). Tuomas O Kilpeläinen was supported by a grant from the Novo Nordisk Foundation (NNF17OC0026848). The funding source had no role in study design, data collection, data analysis, data interpretation, or writing of the report. All authors had full access to all of the data in the study and had final responsibility for the decision to submit for publication.

## Additional information

### Funding

| Funder | Grant reference number | Author |
|---|---|---|
| H2020 Marie Skłodowska-Curie Actions | 846502 | Germán Darío Carrasquilla |
| Novo Nordisk Foundation Center for Basic Metabolic Research | NNF18CC0034900 | Tuomas Kilpeläinen |
| Danish Diabetes Academy | NNF17SA0031406 | Germán Darío Carrasquilla |
| Novo Nordisk Fonden | NNF17OC0026848 | Tuomas Kilpeläinen |

The funders had no role in study design, data collection and interpretation, or the decision to submit the work for publication.

### Author contributions

Germán D Carrasquilla, Conceptualization, Data curation, Formal analysis, Funding acquisition, Methodology, Project administration, Supervision, Visualization, Writing – original draft, Writing – review and editing; Mario García-Ureña, Data curation, Formal analysis, Methodology, Visualization, Writing – review and editing; Tove Fall, Thorkild IA Sørensen, Supervision, Writing – review and editing; Tuomas O Kilpeläinen, Conceptualization, Data curation, Funding acquisition, Methodology, Resources, Supervision, Writing – review and editing

### Author ORCIDs

Germán D Carrasquilla (iD) http://orcid.org/0000-0002-7147-9421
Mario García-Ureña (iD) http://orcid.org/0000-0002-3376-9460
Tove Fall (iD) http://orcid.org/0000-0003-2071-5866
Thorkild IA Sørensen (iD) http://orcid.org/0000-0003-4821-430X
Tuomas O Kilpeläinen (iD) http://orcid.org/0000-0002-8349-3028

### Decision letter and Author response

Decision letter https://doi.org/10.7554/eLife.70386.sa1
Author response https://doi.org/10.7554/eLife.70386.sa2

## Additional files

### Supplementary files

• Transparent reporting form

### Data availability

Data sharing: All analyses were performed using R statistical package freely available at https://cran.r-project.org/mirrors.html. The CAUSE R package and instructions are available at https://jean997.

github.io/cause/. The Two-sample MR package is available at https://mrcieu.github.io/TwoSampleMR/. The RadialMR package is available at https://github.com/WSpiller/RadialMR. The code and curated data for the current analysis are available at https://github.com/MarioGuCBMR/MR_Physical_Activity_BMI, (copy archived at swh:1:rev:b6c3fcd3a97a2d67329ab26b7c061da5941bd4f2).

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

# Appendix 1

Mendelian randomization using the CAUSE, IVW, Egger, weighted median, and weighted mode methods

## Mendelian randomization using the CAUSE method

The CAUSE method performs Mendelian randomization analyses following six different steps, as described in the CAUSE online tutorial (https://jean997.github.io/cause/ldl_cad.html) and the original publication (*Morrison et al., 2020*). The steps are the following:

1. Installing the following versions of these three packages were used: CAUSE v1.0.0, mixsqp v.0.1–97 and ashr v.2.2–32.
2. Filtering data by including variants with imputation quality score INFO >0.7 and minimum allele frequency (MAF) >0.01.
3. Excluding variants from the Major Histocompatibility Complex (MHC) present in chromosome six between the base pairs 26 M and 34 M in build 37.
4. Merging the exposure and outcome GWAS summary level data. Gwas_merge function from CAUSE package was used to identify the variants present in exposure and outcome summary statistics data and to align exposure and the outcome effect sizes to the same allele
5. Calculating nuisance parameters to correct for sample overlap between exposure and outcome GWAS.
6. Using 1000 Genomes CEU population data after using LDshrink as reference panel to clump the variants.
7. Setting the priors for the three model parameters – causal effect, shared effect and q – and calculating their posterior probabilities. For the causal effect and the shared effect, their priors are set automatically to 0, while for q the software allows the user to set the thresholds for the priors. In this case, q priors are set to qalpha = 1 and qbeta = 2.
8. Calculating two models to fit the posterior probabilities: the sharing model, where the causal effect is set to 0, and the causal model, where the posterior probability for the causal effect is calculated.
9. Comparing the two models, sharing and causal, against the null and against each other with expected log pointwise posterior density (ELPD) method to identify which model is the most fitting for the data.

## Mendelian randomization using the IVW, Egger, weighted median, and weighted mode methods

We used the following parameters to interpret the findings:

1. Since the version of TwoSampleMR used in the analysis v0.5.4 removes duplicates by excluding the second instance when introducing data locally, only the variant that presented the same alleles as in the outcome data and with the lowest p-value were kept. In none of the combinations of traits, variants in the MHC were found.
2. To clump variants, the function ld_clump_local from the package ieugwasr v0.1.5 (https://github.com/MRCIEU/ieugwasr) was used using the updated European 1000 Genomes reference panel available in https://github.com/mrcieu/gwasglue.
3. Only variants with MAF >0.01 and INFO > 0.7 were included in the analysis.
4. Only variants that passed the Steiger filtering using the function steiger_filtering from the package TwoSampleMR were used.
5. The no measurement error (NOME) assumption, which assumes no measurement error between the genetic instrument and the exposure, was checked by calculating the mean F-statistic and the variation of the $I^2$ statistic (*Bowden et al., 2019*). The latter can be calculated with the Isq function from TwoSampleMR package.
6. The causal estimates were considered heterogeneous if the P value for Cochran's Q test was <0.05 and $I^2$ was >0.25. Both estimates were calculated using the meta package.
7. An Egger's intercept p-value < 0.05 was considered as evidence of horizontal pleiotropy.

8. To assess whether MR-Egger regression should be applied instead of the standard IVW model, we used the Rucker framework test.
9. To detect individual pleiotropic variants, we used RadialMR's iterative Cochran's Q method following a p value threshold < 0.05. RadialMR presents two functions: ivw_radial or egger_radial, depending on the Cochran's Q, either IVW's Q or Egger's Q', used. The function used was chosen depending on Rucker framework test's result. If Rucker framework presented contradictory results, an iterative version of the Rucker framework (i.e. rucker_jackknife from TwoSampleMR package) was used to assess whether IVW was still chosen as the main model.

To visualize the effects of heterogeneity and horizontal pleiotropy on the results, we generated leave-one-out forest plots and funnel plots. After removing outlier variants detected with RadialMR, we re-run the Mendelian randomization methods and sensitivity tests (*Appendix 1—table 4* and *Appendix 1—table 5*) and re-generated the plots (*Appendix 1—figure 3*), to make sure that the variants introducing horizontal pleiotropy had been removed. The *F*-statistic ranged between 286–6179, indicating that the instrumental variables are strong (F statistic >10) and weak instrument bias may not be an issue. Furthermore, we tested whether the amount of pleiotropy was independent of instrument strength by calculating a variation of the $I^2$ (*Bowden et al., 2019*). We found it to be above 0.90 in all cases. Below we describe and interpret the findings for each combination of traits.

## Vigorous physical activity → BMI

All MR methods except Egger showed negative causal effects, from which only IVW is significant (p < 0.05) (*Appendix 1—table 4*). No outlier extraction was performed since Cochran's Q test was not significant (p = 0.80) and low $I^2$ (0.0%) showed no evidence of heterogeneity. Egger's intercept was not significant (p = 0.30) and Rucker test indicated that IVW model is more fitting for the data (*Appendix 1—table 4*). Consistent with the sensitivity tests, leave-one-out forest plot (*Appendix 1—figure 3*, panel a) showed that the IVW causal effect did not strongly change after the removal of any of the five independent variants used to calculate the causal effects. Funnel plot (*Appendix 1—figure 3*, panel a) showed no signs of asymmetry, which is consistent with Egger's intercept result. Thus, these findings indicate that the causal effects are not biased and they reflect weak evidence of causality between an increase of vigorous physical activity and a decrease of BMI. While IVW negative causal effect is significantly different from 0, the evidence for causality is considered weak due to (1) IVW's p-value is still close to nominal threshold (p = 0.04), (2) the other methods with negative causal effects are not significantly different from 0 (p > 0.05) and (3) Egger's regression presents a positive causal effect. These weak results are most probably due to the low number of variants used (*Appendix 1—table 4*).

## Moderate physical activity → BMI

All MR methods except Egger showed negative effects directions, with only IVW's being significant (*Appendix 1—table 4*). No outlier extraction was performed since Cochran's Q test was not significant (p = 0.30) and $I^2$ was below 25% (18.9%). Egger's intercept was not significantly different from 0 (p = 0.16) indicating that no pleiotropic variants were found among the four independent variants used to calculate the causal effects. Rucker test indicated that IVW was a better fit for the model than Egger, in line with Egger's intercept results (*Appendix 1—table 4*). The leave-one-out forest sensitivity plot (*Appendix 1—figure 3*, panel c) showed that no variant presented heterogenic causal effect since the IVW causal effect did not strongly diverge after removing any of the variants. The funnel sensitivity plot (*Appendix 1—figure 3*, panel c) showed no asymmetry. All in all, the findings indicate that the causal effects are not biased, but that they suggest causality between increased moderate physical activity and decreased BMI. The reason behind this is that (1) IVW's negative causal effect present a p-value close to the nominal threshold (p = 0.02), (2) the other methods present negative causal effect that are not significantly different from 0 (p > 0.05) and (3) Egger's regression presents a positive causal effect, although it is not significantly different from 0 (*Appendix 1—table 4*).

## Sedentary time → BMI

Nine independent variants were used as instruments, and two were removed after outlier extraction as indicated by RadialMR. All methods were not significant, and all reported a positive causal effect (p > 0.05), except weighted mode (*Appendix 1—table 4*). After outlier extraction, Cochran's Q test was non-significant (p = 0.15), but the $I^2$ was 36.4%, indicating slight heterogeneity. Egger's intercept was not significant (p = 0.43) and Rucker test indicated that IVW method is a better fit for

the data than Egger, implying that pleiotropic variants were not present (*Appendix 1—table 4*). In accordance, the forest plot and funnel plot (*Appendix 1—figure 3*, panel e) showed that the results are not strongly affected by horizontal pleiotropy. The forest plot indicated that the heterogeneity comes from variants with both positive and negative causal effects, since the IVW causal effect becomes either more strongly negative or positive when the variants are removed one at a time. The funnel plot (*Appendix 1—figure 3*, panel e) showed that the causal estimates are evenly distributed, implying no asymmetry and hence no pleiotropy. To conclude, the findings indicate a non-significant positive causal effect between sedentary time and BMI.

### BMI → vigorous physical activity

All MR methods except Egger presented negative causal estimates. The causal estimate from the IVW method reached statistical significance ($p < 0.01$) (*Appendix 1—table 5*). Non-significant p value from Cochran's Q test ($p = 0.27$) and low $I^2$ of 9.6% indicated no heterogeneity. Egger intercept's was significant ($p = 0.0017$) and IVW's Q and Egger's Q were significantly different (Q-Q' = 9.83, $p = 0.0017$), indicating that horizontal pleiotropy may still affect the causal estimates. Rucker framework selected IVW more fitting for the data than Egger (*Appendix 1—table 5*). Forest and funnel plots (*Appendix 1—figure 3*, panel b) indicated that some variants, both with positive and negative effects, may be introducing heterogeneity. The forest plot showed slight deviations from the mean causal effect when extreme variants were extracted one at a time. In the funnel plot, symmetry was disrupted by variants with positive causal effects and small standard errors. Considering both plots, we infer that heterogeneity and horizontal pleiotropy were introduced by specific variants with small standard errors. Leave-one-out plots indicated that two variants (rs13021737 and rs6567160) introduced horizontal pleiotropy, showing positive causal effects and having the smallest standard errors of all variants. To conclude, we found residual pleiotropy that RadialMR could not properly address and deem that the association between BMI and vigorous physical activity is unlikely to be causal.

### BMI → moderate physical activity

The IVW method showed a negative causal effect, whereas Egger, weighted median and weighted mode showed a positive causal effect (*Appendix 1—table 5*). Only the causal effect from Egger was significant ($p = 0.04$). Non-significant p from Cochran's Q test ($p = 0.91$) and $I^2$ of 0% indicated no presence of heterogeneity. Egger intercept's was significant ($P = 0.002$) and IVW's Q and Egger's Q were significantly different (Q-Q' = 9.60, $p = 1.95 \times 10^{-5}$), indicating that horizontal pleiotropy may still affect the causal estimates. Rucker framework chose IVW as the most fitting method for the data. Forest and funnel plots (*Appendix 1—figure 3*, panel d) indicated heterogeneity. The leave-one-out forest plots indicated deviation from the mean causal effect when variants with positive causal effects were excluded, and the funnel plots showed that asymmetry was introduced by variants with positive causal effects and small standard errors. These findings were in agreement with the findings from Rucker framework and Egger's intercept test results. To conclude, we found evidence of horizontal pleiotropy that RadialMR could not properly remove, and where pleiotropy was introduced by variants with positive causal estimates. The presence of horizontal pleiotropy may explain the negative direction of the causal estimate in the IVW method, whereas other methods showed positive causal estimates (*Appendix 1—table 5*). We deem that the association between BMI and moderate physical activity is unlikely to be causal.

### BMI → sedentary time

The IVW and weighted median methods showed positive causal effects, while Egger and weighted mode presented negative causal effects (*Appendix 1—table 5*), with only IVW being significant ($p = 0.02$). The Cochran's Q test ($p = 0.4$) and $I^2$ of 2.2% indicated no presence of heterogeneity. Egger intercept test ($p = 0.20$) and the non-significant difference between IVW's Q and Egger's Q (Q-Q' = 1.62, $p = 0.20$) indicated no presence of horizontal pleiotropy and that the IVW method was a better fit for the data than Egger (*Appendix 1—table 5*). Forest and funnel sensitivity plots (*Appendix 1—figure 3*, panel f) indicated that there was no heterogeneity, except for one variant (rs6567160) with a negative effect. In the forest plot, upper and lower extreme variants induced deviations from the mean causal effect when extracted one at a time. Taken together, sensitivity tests indicate no horizontal pleiotropy and forest and funnel plots indicate the presence of heterogeneity for one variant, while RadialMR could not detect any outliers. To conclude, as the forest and funnel plots indicated heterogeneity, implying that there was residual horizontal pleiotropy, a causal relationship between BMI and sedentary time cannot be deemed, despite the significant p value in the IVW method.

### Conversion of the causal estimates to absolute units

To interpret the causal estimates from CAUSE, we applied the weighted median method on the independent instrumental variants selected by the CAUSE method and performed outlier extraction and sensitivity tests, following the steps described below. The results are reported in *Appendix 1—table 3*.

1. We selected independent variants associated with each exposure identified with the CAUSE method.
2. We performed outlier extraction using RadialMR to control for uncorrelated pleiotropy.
3. We obtained weighted median's causal estimates, to correct for correlated pleiotropy.
4. We approximated the causal estimates between outcome and exposure in absolute units. As the genome-wide summary results were reported in standard deviation units, we multiplied the causal estimates by the standard deviation of the corresponding non-transformed trait to derive the estimates in original trait units. The equations used in these calculations are described for each combination of traits in *Appendix 1—table 3*.

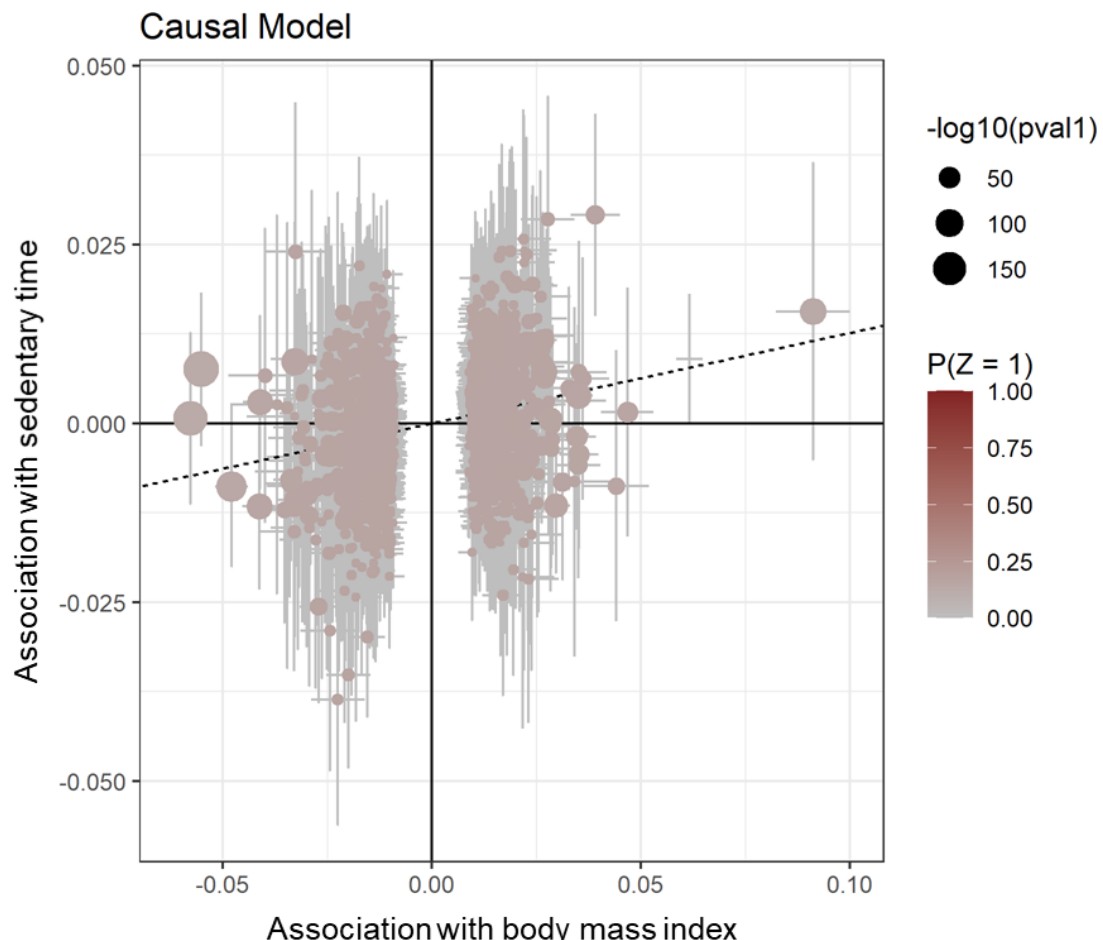

**Appendix 1—figure 1.** Scatter plot for CAUSE test statistics on the causal association between BMI and sedentary time. Effect estimates for body mass index (x-axis) are plotted against estimates for sedentary time (y-axis). Error bars indicate 95% confidence intervals and the dotted line indicates the median causal effect of BMI on sedentary time.

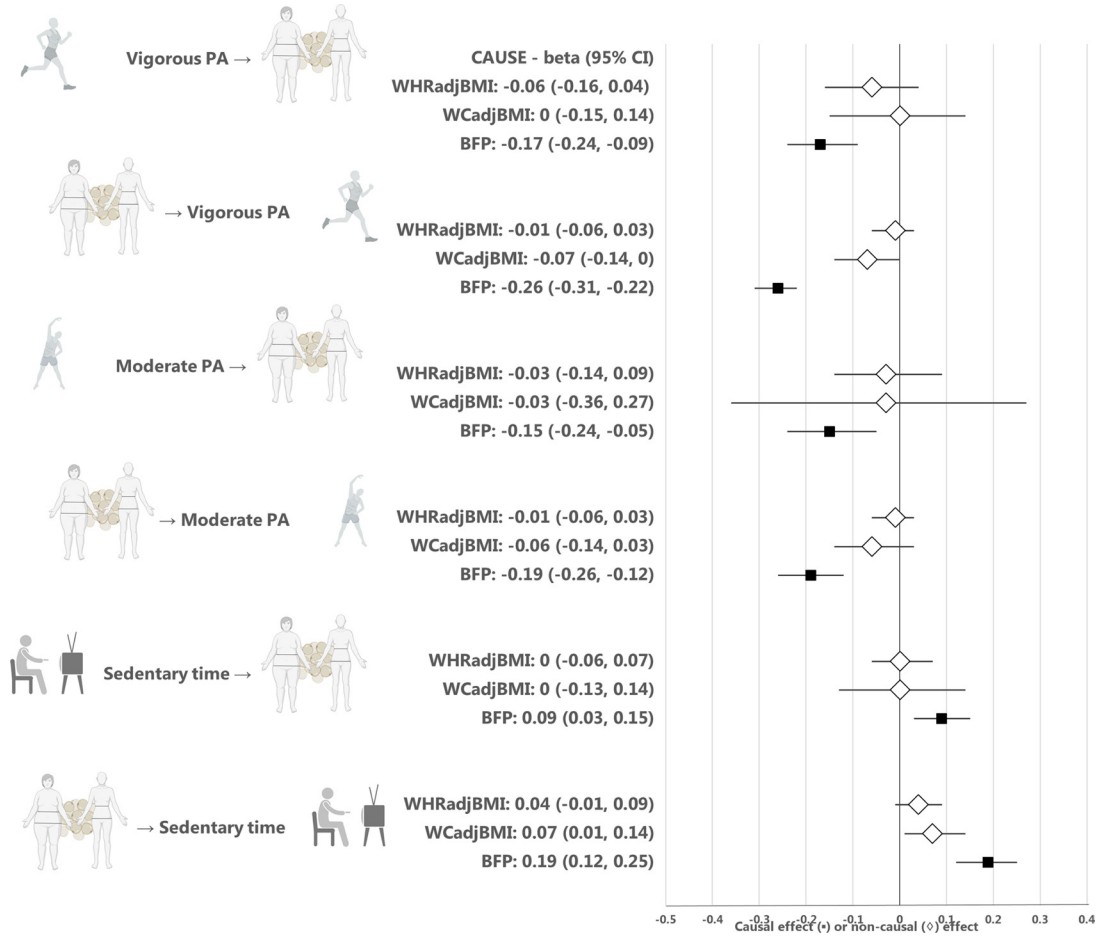

**Appendix 1—figure 2.** Estimates from the CAUSE method for the causal relationship between physical activity or sedentary time and waist-hip ratio, waist circumference, or body fat percentage. Median causal estimates for Mendelian randomization analyses using the CAUSE method. A diamond (◊) in the estimate for CAUSE indicates that the sharing model fit the data better than the causal model, that is that the association between the traits was more likely to be explained by horizontal correlated pleiotropy than causality. PA, physical activity; WHRadjBMI, waist-to-hip ratio adjusted for body mass index; WCadjBMI, waist circumference adjusted for body mass index; BFP, body fat percentage. Figure icons were created with BioRender.com.

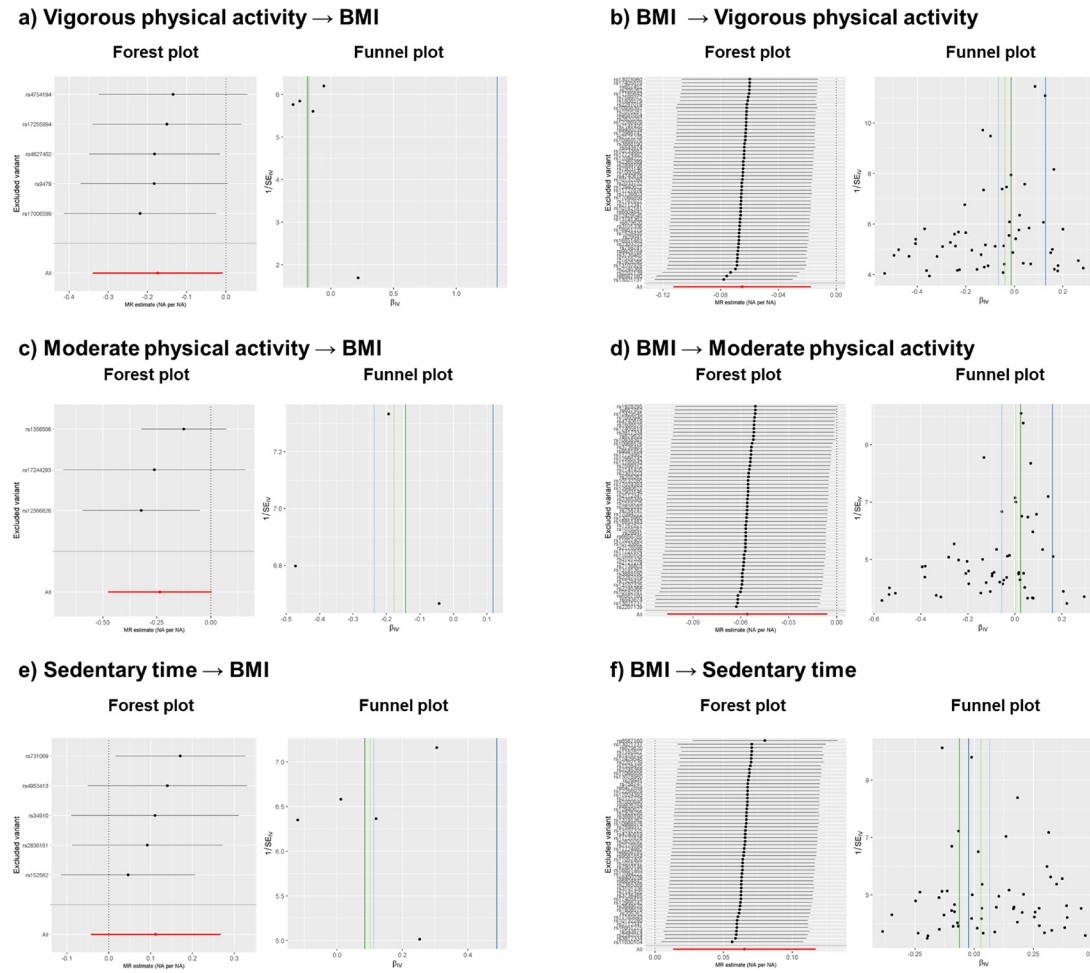

**Appendix 1—figure 3.** Leave-one-out forest and funnel sensitivity plots after outlier extraction.

**Appendix 1—table 1.** CAUSE expected log pointwise posterior density (ELPD) results for each combination of traits.

| Model 1 | Model 2 | Vigorous physical activity → BMI | | | BMI → vigorous physical activity | | |
|---|---|---|---|---|---|---|---|
| | | Delta_ELPD | SE_delta_ELPD | z | Delta_ELPD | SE_delta_ELPD | Z |
| Null | Sharing | −17.47 | 4.24 | −4.12 | −251.88 | 21.54 | −11.69 |
| Null | Causal | −18.98 | 4.68 | −4.05 | −252.22 | 21.65 | −11.65 |
| Sharing | Causal | −1.51 | 0.45 | −3.36 | −0.35 | 0.90 | −0.38 |

| | | Moderate physical activity → BMI | | | BMI → Moderate physical activity | | |
|---|---|---|---|---|---|---|---|
| Null | Sharing | −8.50 | 2.86 | −2.97 | −119.39 | 15.15 | −7.88 |
| Null | Causal | −9.83 | 3.39 | −2.90 | −119.77 | 15.34 | −7.81 |
| Sharing | Causal | −1.34 | 0.53 | −2.52 | −0.38 | 0.77 | −0.49 |

| | | Sedentary time → BMI | | | BMI → Sedentary time | | |
|---|---|---|---|---|---|---|---|
| Null | Sharing | −6.51 | 2.57 | −2.53 | −137.84 | 157.46 | −8.75 |
| Null | Causal | −7.67 | 3.14 | −2.44 | −138.89 | 15.88 | −8.75 |
| Sharing | Causal | −1.16 | 0.57 | −2.05 | −1.06 | 0.33 | −3.22 |

BMI, Body mass index; SE, standard error; NA, not applicable; CI, 95% confidence interval.

**Appendix 1—table 2.** CAUSE posterior probabilities and q values for the causal effect and the shared effect.

| Model | Causal effect (CI) | Shared effect (CI) | q (CI) | Sharing vs causal model p-value |
|---|---|---|---|---|
| | **Vigorous physical activity → BMI** | | | |
| Sharing | NA | –0.18 (-0.25,–0.12) | 0.85 (0.57, 0.98) | |
| Causal | –0.16 (-0.24,–0.08) | 0 (-0.31, 0.31) | 0.19 (0.00, 0.86) | 3.80E-04 |
| | **Moderate physical activity → BMI** | | | |
| Sharing | NA | –0.2 (-0.33,–0.11) | 0.75 (0.35, 0.96) | |
| Causal | –0.18 (-0.3,–0.05) | –0.01 (-0.44, 0.39) | 0.2 (0.01, 0.86) | 5.80E-03 |
| | **Sedentary time → BMI** | | | |
| Sharing | NA | 0.13 (0.06, 0.23) | 0.69 (0.25, 0.95) | |
| Causal | 0.11 (0.02, 0.20) | 0 (-0.32, 0.32) | 0.19 (0, 0.86) | 2.00E-02 |
| | **BMI → Vigorous physical activity** | | | |
| Sharing | NA | –0.16 (-0.19,–0.14) | 0.9 (0.77, 0.98) | |
| Causal | –0.15 (-0.2,–0.08) | 0.01 (-0.18, 0.21) | 0.21 (0.01, 0.83) | 0.35 |
| | **BMI → Moderate physical activity** | | | |
| Sharing | NA | –0.14 (-0.19,–0.11) | 0.77 (0.55, 0.95) | |
| Causal | –0.11 (-0.19,–0.03) | 0 (-0.28, 0.2) | 0.23 (0.01, 0.84) | 0.31 |
| | **BMI → Sedentary time** | | | |
| Sharing | NA | 0.14 (0.12, 0.17) | 0.9 (0.73, 0.98) | |
| Causal | 0.13 (0.08, 0.17) | 0 (-0.23, 0.26) | 0.18 (0.00, 0.85) | 6.30E-04 |

BMI, Body mass index; NA, not applicable; CI, 95% confidence interval.

**Appendix 1—table 3.** Approximation of the causal estimates in absolute units.

| Analysis | Original causal estimates (weighted median) | | | | Conversion of causal estimates | | | Converted causal estimates |
|---|---|---|---|---|---|---|---|---|
| | N SNPs | Beta | SE | p | Equation | SDexposure | SDoutcome | |
| Vigorous physical activity → BMI | 1,036 | –0.09 SD | 0.006 SD | 9.28E-57 | (Beta*SDoutcome)/ SDexposure / (0,01*24 hours) | NA | 4·807 kg/m2 | –0·48 kg/m2 per hour of moderate physical activity per day |
| Moderate physical activity → BMI | 914 | –0.05 SD | 0.005 SD | 1.21E-20 | (Beta*SDoutcome)/ SDexposure / (0,01*24 hours) | 3.76% | 4·807 kg/m2 | –0·27 kg/m2 per hour of moderate physical activity per day |
| Sedentary time → BMI | 1,036 | 0·06 SD | 0.005 SD | 6.33E-30 | (Beta*SDoutcome)/ SDexposure / (0,01*24 hours) | 7.38% | 4·807 kg/m2 | 0,14 kg/m2 per hour of sedentary time per day |
| BMI → Sedentary time | 11,849 | 0·16 SD | 0·009 SD | 2.87E-68 | (Beta*SDoutcome)/ SDexposure *(0·01*24 hours) | 4·807 kg/m2 | 7.38% | 3·54 min of sedentary time per kg/m2 per day |

Original units of the traits:
Moderate physical activity: Probability of moderate physical activity in a 30s-epoch frame (*Doherty et al., 2018*)
Sedentary time: Probability of sedentary time in a 30s-epoch frame (*Doherty et al., 2018*)
BMI: kg/m² (*Pulit et al., 2019*)
Note: For vigorous physical activity, SD exposure could not be found, thus we use the same as for moderate physical activity, i.e. 3·76%

SNP, single nucleotide polymorphism; BMI, SE, standard error; SD, standard deviation; Body mass index (BMI).

**Appendix 1—table 4.** Mendelian randomization results using the IVW, Egger, weighted median and weighted mode methods of for the causal effect of moderate or vigorous physical activity or sedentary time on BMI.

**Before outlier extraction**

| MR method | Vigorous physical activity → BMI | | | | Moderate physical activity → BMI | | | | Sedentary time → BMI | | | |
|---|---|---|---|---|---|---|---|---|---|---|---|---|
| | SNP | beta | SE | p-value | SNP | beta | SE | p-value | SNP | beta | SE | p-value |
| Egger | 5 | 1.33 | 2.21 | 0.59 | 3 | 0.12 | 0.45 | 0.84 | 8 | 0.46 | 0.64 | 0.50 |
| Weighted median | 5 | –0.18 | 0.10 | 0.08 | 3 | –0.18 | 0.12 | 0.13 | 8 | 0.02 | 0.09 | 0.82 |
| IVW | 5 | –0.17 | 0.08 | 0.04 | 3 | –0.24 | 0.12 | 0.05 | 8 | 0.04 | 0.11 | 0.72 |
| Weighted mode | 5 | –0.19 | 0.12 | 0.19 | 3 | –0.14 | 0.15 | 0.43 | 8 | –0.05 | 0.18 | 0.78 |

| Sensitivity test | Estimate | p-value/CI | Estimate | p-value/CI | Estimate | p-value/CI |
|---|---|---|---|---|---|---|
| Q (p-value) | 1.65 | 0.8 | 3.6 | 0.17 | 23.08 | 1.70E-03 |
| I2 (CI) | 0.00% | (0,0%; 49,5%) | 44.30% | (0,0%; 83,4%) | 69.70% | (36,8%; 85,4%) |
| Q-Q' (p-value) | 0.46 | 0.5 | 1.75 | 0.19 | 1.96 | 0.16 |
| Egger intercept (p-value) | –0.04 | 0.29 | –0.01 | 0.19 | –0.01 | 0.75 |
| Rucker Test | IVW | | IVW | | IVW | |

**After outlier extraction**

| MR method | Vigorous physical activity → BMI | | | | Moderate physical activity → BMI | | | | Sedentary time → BMI | | | |
|---|---|---|---|---|---|---|---|---|---|---|---|---|
| | SNP | beta | SE | p-value | SNP | beta | SE | p-value | SNP | beta | SE | p-value |
| Egger | NA | NA | NA | NA | NA | NA | NA | NA | 5 | 0.49 | 0.36 | 0.26 |
| Weighted median | NA | NA | NA | NA | NA | NA | NA | NA | 5 | 0.10 | 0.10 | 0.29 |
| IVW | NA | NA | NA | NA | NA | NA | NA | NA | 5 | 0.11 | 0.08 | 0.16 |
| Weighted mode | NA | NA | NA | NA | NA | NA | NA | NA | 5 | 0.08 | 0.14 | 0.59 |

| Sensitivity test | Estimate | p-value/CI | Estimate | p-value/CI | Estimate | p-value/CI |
|---|---|---|---|---|---|---|
| Q (p-value) | NA | NA | NA | NA | 4.7 | 0.32 |
| I2 (CI) | NA | NA | NA | NA | 15 | (0·0%; 82·3%) |
| Q-Q' (p-value) | NA | NA | NA | NA | 1.41 | 0.24 |
| Egger intercept (p-value) | NA | NA | NA | NA | –1.00E-02 | 0.24 |
| Rucker Test | NA | | NA | | IVW | |

BMI, Body mass index (BMI); MR, Mendelian randomization; SNP, single nucleotide polymorphism; SE, standard error; NA, not applicable; IVW, inverse variance weighted; CI, 95% confidence interval. NA: not applicable. Note: No outlier extraction was performed for moderate physical activity → BMI and vigorous physical activity → BMI directions, since Q test and I2 did not indicate heterogeneity.

**Appendix 1—table 5.** Mendelian randomization results using the IVW, Egger, weighted median and weighted mode methods of BMI on moderate PA, vigorous PA, or sedentary time.

**Before outlier extraction**

| MR method | BMI → Vigorous physical activity | | | | BMI → Moderate physical activity | | | | BMI → Sedentary time | | | |
|---|---|---|---|---|---|---|---|---|---|---|---|---|
| | SNP | beta | SE | p-value | SNP | beta | SE | p-value | SNP | beta | SE | p-value |
| Egger | 64 | 0.09 | 0.08 | 0.22 | 65 | 0.17 | 0.08 | 0.04 | 65 | –0.10 | 0.09 | 0.28 |
| Weighted median | 64 | –0.04 | 0.03 | 0.21 | 65 | 0.004 | 0.04 | 0.92 | 65 | 0.03 | 0.04 | 0.44 |
| IVW | 64 | –0.08 | 0.03 | 0.005 | 65 | –0.05 | 0.03 | 0.09 | 65 | 0.07 | 0.03 | 0.02 |
| Weighted mode | 64 | –0.01 | 0.07 | 0.86 | 65 | 0.02 | 0.05 | 0.72 | 65 | –0.05 | 0.10 | 0.59 |

| Sensitivity test | Estimate | p-value/CI | Estimate | p-value/CI | Estimate | p-value/CI |
|---|---|---|---|---|---|---|
| Q (p-value) | 92.4 | 9.30E-03 | 83.73 | 0.05 | 97.5 | 4.40E-03 |
| I2 (CI) | 31.80% | 6·9%; 50·00% | 23.60% | (0·00%; 44·20%) | 34.40% | 10·80%; 51·70% |

*Continued on next page*

*Continued*

| Sensitivity test | Estimate | p-value/CI | Estimate | p-value/CI | Estimate | p-value/CI |
|---|---|---|---|---|---|---|
| Q-Q' (p-value) | 8.36 | 3.80E-03 | 10.86 | 9.81E-04 | 6.59 | 1.00E-02 |
| Egger intercept (p-value) | −4.99E-03 | 9.90E-01 | −6.40E-03 | 9.81E-04 | 4.96E-03 | 2.00E-02 |
| Rucker test | | Egger | | Egger | | Egger |

**After outlier extraction**

| | BMI → Vigorous physical activity | | | | BMI → Moderate physical activity | | | | BMI → Sedentary time | | | |
|---|---|---|---|---|---|---|---|---|---|---|---|---|
| MR method | SNP | beta | SE | p-value | SNP | beta | SE | p-value | SNP | beta | SE | p-value |
| Egger | 57 | 0.13 | 0.07 | 0.06 | 55 | 0.16 | 0.07 | 0.04 | 57 | −0.02 | 0.08 | 0.76 |
| Weighted median | 57 | −0.04 | 0.03 | 0.27 | 55 | 2.95E-03 | 0.04 | 0.93 | 57 | 0.03 | 0.04 | 0.48 |
| IVW | 57 | −0.07 | 0.02 | 0.01 | 55 | −0.06 | 0.03 | 0.03 | 57 | 0.07 | 0.03 | 0.01 |
| Weighted mode | 57 | −0.01 | 0.06 | 0.83 | 55 | 0.02 | 0.06 | 0.68 | 57 | −0.06 | 0.08 | 0.46 |

| Sensitivity test | Estimate | p-value/CI | Estimate | p-value/CI | Estimate | p-value/CI |
|---|---|---|---|---|---|---|
| Q (p-value) | 61.88 | 0.27 | 40.51 | 0.91 | 56.29 | 0.46 |
| I2 (CI) | 9.50% | 0·00%–35·10% | 0.00% | (0·0%; 8·9%) | 0.50% | 0·0%; 31·50% |
| Q-Q' (p-value) | 9.79 | 1.75E-03 | 9.6 | 1.94E-03 | 1.62 | 2.00E-01 |
| Egger intercept | −5.70E-03 | 1.75E-03 | −6.44E-03 | 9.15E-05 | 2.60E-03 | 2.00E-01 |
| Rucker test | | IVW | | IVW | | IVW |

BMI, Body mass index; MR, Mendelian randomization; SNP, single nucleotide polymorphism; SE, standard error; NA, not applicable; IVW, inverse variance weighted; CI, 95% confidence interval.

**Appendix 1—table 6.** CAUSE expected log pointwise posterior density (ELPD) results for central adiposity and fat percentage.

| | | Vigorous physical activity → BFP | | | BFP → vigorous physical activity | | |
|---|---|---|---|---|---|---|---|
| Model 1 | Model 2 | Delta_ELPD | SE_delta_ELPD | z | Delta_ELPD | SE_delta_ELPD | z |
| Null | Sharing | −23.55 | 4.67 | −5.04 | −360.48 | 24.46 | −14.74 |
| Null | Causal | −25.11 | 5.00 | −5.02 | −361.77 | 24.49 | −14.77 |
| Sharing | Causal | −1.56 | 0.33 | −4.68 | −1.30 | 0.17 | −7.77 |

| | | Moderate physical activity → BFP | | | BFP → Moderate physical activity | | |
|---|---|---|---|---|---|---|---|
| Null | Sharing | −8.71 | 2.60 | −3.34 | −170.46 | 17.46 | −9.76 |
| Null | Causal | −10.11 | 3.13 | −3.23 | −171.50 | 17.63 | −9.73 |
| Sharing | Causal | −1.40 | 0.53 | −2.65 | −1.03 | 0.54 | −1.90 |

| | | Sedentary time → BFP | | | BFP → Sedentary time | | |
|---|---|---|---|---|---|---|---|
| Null | Sharing | −9.74 | 3.30 | −2.95 | −143.04 | 15.86 | −9.02 |
| Null | Causal | −11.01 | 3.83 | −2.87 | −144.25 | 16.13 | −8.94 |
| Sharing | Causal | −1.27 | 0.54 | −2.38 | −1.21 | 0.36 | −3.41 |

| | | Vigorous physical activity → WCadjBMI | | | WCadjBMI → Vigorous physical activity | | |
|---|---|---|---|---|---|---|---|
| | | Delta_ELPD | SE_delta_ELPD | z | Delta_ELPD | SE_delta_ELPD | z |
| Null | Sharing | 0.57 | 0.07 | 7.71 | −6.37 | 3.13 | −2.03 |
| Null | Causal | 1.36 | 0.15 | 9.28 | −7.16 | 3.65 | −1.96 |
| Sharing | Causal | 0.79 | 0.07 | 10.60 | −0.79 | 0.54 | −1.46 |

| | | Moderate physical activity → WCadjBMI | | | WCadjBMI → Moderate physical activity | | |
|---|---|---|---|---|---|---|---|
| Null | Sharing | 0.50 | 0.56 | 0.89 | –4.83 | 2.96 | –1.63 |
| Null | Causal | 1.01 | 0.87 | 1.16 | –5.40 | 3.58 | –1.51 |
| Sharing | Causal | 0.51 | 0.36 | 1.44 | –0.56 | 0.65 | –0.87 |

| | | Sedentary time → WCadjBMI | | | WCadjBMI → Sedentary time | | |
|---|---|---|---|---|---|---|---|
| Null | Sharing | 0.55 | 0.07 | 7.56 | –7.59 | 3.34 | –2.27 |
| Null | Causal | 1.39 | 0.16 | 8.55 | –8.61 | 3.97 | –2.17 |
| Sharing | Causal | 0.83 | 0.09 | 9.21 | –1.02 | 0.65 | –1.56 |

| | | Vigorous physical activity → WHRadjBMI | | | WHRadjBMI → Vigorous physical activity | | |
|---|---|---|---|---|---|---|---|
| | | Delta_ELPD | SE_delta_ELPD | z | Delta_ELPD | SE_delta_ELPD | z |
| Null | Sharing | –0.91 | 1.19 | –0.77 | 0.06 | 0.84 | 0.07 |
| Null | Causal | –1.68 | 2.10 | –0.80 | 0.31 | 1.52 | 0.21 |
| Sharing | Causal | –0.77 | 0.91 | –0.85 | 0.26 | 0.68 | 0.37 |

| | | Moderate physical activity → WHRadjBMI | | | WHRadjBMI → Moderate physical activity | | |
|---|---|---|---|---|---|---|---|
| Null | Sharing | 0.48 | 0.16 | 3.00 | 0.22 | 0.77 | 0.28 |
| Null | Causal | 0.92 | 0.56 | 1.63 | 0.44 | 1.44 | 0.30 |
| Sharing | Causal | 0.44 | 0.41 | 1.09 | 0.22 | 0.67 | 0.32 |

| | | Sedentary time → WHRadjBMI | | | WHRadjBMI → Sedentary time | | |
|---|---|---|---|---|---|---|---|
| Null | Sharing | 0.57 | 0.11 | 5.29 | –6.72 | 3.44 | –1.95 |
| Null | Causal | 1.37 | 0.28 | 4.83 | –7.31 | 4.03 | –1.81 |
| Sharing | Causal | 0.80 | 0.18 | 4.48 | –0.59 | 0.61 | –0.97 |

BFP, body fat percentage; WCadjBMI, waist cirumference adjusted for body mass index; WHRadjBMI, waist-to-hip ratio adjusted for body mass index, SE, standard error; NA, not applicable; CI, 95% confidence interval.

**Appendix 1—table 7.** CAUSE posterior probabilities and q values for the causal and sharing models for central adiposity and body fat percentage.

| Model | Causal Effect (CI) | Shared effect (CI) | Q (CI) | Sharing vs causal model p-value |
|---|---|---|---|---|
| | Vigorous physical activity → BFP | | | |
| Sharing | NA | –0.18 (-0.23,–0.13) | 0.89 (0.67, 0.98) | |
| Causal | –0.17 (-0.24,–0.09) | 0 (-0.28, 0.26) | 0.19 (0, 0.86) | 1.4e-06 |
| | Moderate physical activity → BFP | | | |
| Sharing | NA | –0.17 (-0.26,–0.09) | 0.78 (0.39, 0.97) | |
| Causal | –0.15 (-0.24,–0.05) | 0 (-0.32, 0.31) | 0.2 (0, 0.86) | 0.0041 |
| | Sedentary time → BFP | | | |
| Sharing | NA | 0.11 (0.06, 0.17) | 0.75 (0.37, 0.96) | |
| Causal | 0.09 (0.03, 0.15) | 0 (-0.26, 0.27) | 0.19 (0, 0.86) | 0.0088 |

| | | | | |
|---|---|---|---|---|
| **BFP → Vigorous physical activity** | | | | |
| Sharing | NA | –0.27 (-0.29,–0.25) | 0.97 (0.92, 1) | |
| Causal | –0.26 (-0.31,–0.22) | 0 (-0.26, 0.22) | 0.18 (0, 0.85) | 3.9e-15 |
| BFP → Moderate physical activity | | | | |
| Sharing | NA | –0.21 (-0.25,–0.18) | 0.91 (0.77, 0.99) | |
| Causal | –0.19 (-0.26,–0.12) | 0 (-0.3, 0.31) | 0.19 (0, 0.86) | 0.029 |
| **BFP → Sedentary time** | | | | |
| Sharing | NA | 0.2 (0.17, 0.24) | 0.91 (0.76, 0.99) | |
| Causal | 0.19 (0.12, 0.25) | 0 (-0.33, 0.31) | 0.18 (0, 0.86) | 0.00032 |
| | | | | |
| **Vigorous physical activity → WCadjBMI** | | | | |
| Sharing | NA | –0.01 (-0.62, 0.59) | 0.12 (0, 0.72) | |
| Causal | 0 (-0.15, 0.14) | 0 (-0.55, 0.52) | 0.2 (0.01, 0.86) | 1 |
| **Moderate physical activity → WCadjBMI** | | | | |
| Sharing | NA | –0.14 (-0.98, 0.6) | 0.13 (0.01, 0.69) | |
| Causal | –0.03 (-0.36, 0.27) | –0.03 (-0.83, 0.7) | 0.26 (0.01, 0.84) | 9.20E-01 |
| **Sedentary time → WCadjBMI** | | | | |
| Sharing | NA | 0.01 (-0.44, 0.57) | 0.12 (0, 0.71) | |
| Causal | 0 (-0.13, 0.14) | 0.01 (-0.41, 0.52) | 0.21 (0.01, 0.85) | 1 |
| **WCadjBMI → Vigorous physical activity** | | | | |
| Sharing | NA | –0.1 (-0.21,–0.05) | 0.59 (0.19, 0.92) | |
| Causal | –0.07 (-0.14, 0) | 0 (-0.31, 0.36) | 0.19 (0, 0.86) | 0.072 |
| **WCadjBMI → Moderate physical activity** | | | | |
| Sharing | NA | –0.11 (-0.27,–0.04) | 0.47 (0.11, 0.88) | |
| Causal | –0.06 (-0.14, 0.03) | 0 (-0.34, 0.36) | 0.19 (0, 0.86) | 0.19 |
| **WCadjBMI → Sedentary time** | | | | |
| Sharing | NA | 0.1 (0.05, 0.2) | 0.63 (0.23, 0.93) | |
| Causal | 0.07 (0.01, 0.14) | 0.01 (-0.3, 0.48) | 0.17 (0, 0.85) | 0.059 |
| | | | | |
| **Vigorous physical activity → WHRadjBMI** | | | | |
| Sharing | NA | –0.1 (-0.39, 0.07) | 0.36 (0.01, 0.86) | |
| Causal | –0.06 (-0.16, 0.04) | 0 (-0.37, 0.44) | 0.19 (0, 0.86) | 0.2 |
| **Moderate physical activity → WHRadjBMI** | | | | |
| Sharing | NA | –0.04 (-0.48, 0.39) | 0.15 (0, 0.76) | |
| Causal | –0.03 (-0.14, 0.09) | 0 (-0.41, 0.4) | 0.2 (0, 0.86) | 8.60E-01 |
| **Sedentary time → WHRadjBMI** | | | | |
| Sharing | NA | 0.01 (-0.26, 0.34) | 0.12 (0, 0.72) | |
| Causal | 0 (-0.06, 0.07) | 0 (-0.22, 0.28) | 0.2 (0, 0.86) | 1 |

| | | | | |
|---|---|---|---|---|
| **WHRadjBMI → Vigorous physical activity** | | | | |
| Sharing | NA | –0.04 (-0.3, 0.12) | 0.16 (0, 0.76) | |
| Causal | –0.01 (-0.06, 0.03) | 0 (-0.25, 0.21) | 0.18 (0, 0.85) | 0.65 |
| **WHRadjBMI → Moderate physical activity** | | | | |
| Sharing | NA | –0.04 (-0.31, 0.14) | 0.16 (0, 0.75) | |
| Causal | –0.01 (-0.06, 0.03) | 0 (-0.25, 0.22) | 0.18 (0, 0.85) | 6.30E-01 |
| **WHRadjBMI → Sedentary time** | | | | |
| Sharing | NA | 0.07 (0.03, 0.18) | 0.49 (0.13, 0.89) | |
| Causal | 0.04 (-0.01, 0.09) | 0 (-0.29, 0.4) | 0.17 (0, 0.83) | 0.17 |

BFP, Body Fat Percentage; WCadjBMI, waist circumference adjusted for BMI; WHRadjBMI, waist-to-hip ratio adjusted for BMI; NA, not applicable; CI, 95% confidence interval.

## Appendix 2

### Analysis plan for causality between physical activity, sedentary behaviour, and obesity: A Mendelian randomization study

Date: initiated in March 2020

Motivation: Obesity is a global epidemic increasing morbidity and mortality worldwide. Physical inactivity and increased sedentary time are associated with excess weight gain as shown from observational studies. However, observational studies suffer from residual confounding and reverse causality. Mendelian randomization helps to overcome confounding and reverse causality by instrumenting the exposure trait using genetic variants. Here, we aim to assess the causality between physical activity, sedentary behaviour and body mass index in adults by bidirectional Mendelian randomization analyses.

Inclusion criteria for genome wide summary level data:

- Largest published genome-wide association study summary statistics
- European ancestry
- Objectively measured continuous traits

Exposures:

- Vigorous physical activity
- Moderate physical activity
- Sedentary time
- Body mass index

Outcomes:

- Vigorous physical activity
- Moderate physical activity
- Sedentary time
- Body mass index

Directions of association for outcome and exposure:

- Vigorous physical activity → body mass index
- Moderate physical activity → body mass index
- Sedentary time → body mass index
- Body mass index → vigorous physical activity
- Body mass index → moderate physical activity
- Body mass index → sedentary time

Mendelian randomization methods:

- The Causal Analysis Using Summary Effect estimates (CAUSE) method
- Inverse variance weighted (IVW)
- Mendelian randomization-Egger (MR-Egger)
- Weighted median
- Weighted mode

Sensitivity tests and plots to account for heterogeneity and horizontal pleiotropy in the IVW, MR-Egger, weighted median and weighted mode analyses:

- Steiger filtering
- Automated outlier removal with RadialMR
- Rucker framework
- Cochran's Q method
- Leave-one-out forest plots
- Funnel plots

Note: The Mendelian randomization analyses, sensitivity tests, and sensitivity plots for IVW, MR-Egger, weighted median and weighted mode methods were repeated after outlier removal.

Post-hoc analysis:

When less than three genetic variants associated with the exposure trait were available for the IVW, MR-Egger, weighted median, and weighted mode methods using genome-wide significant threshold ($p < 5 \times 10^{-8}$), we used a p-value threshold of $p < 5 \times 10^{-7}$ to identify a sufficient number of genetic instruments to produce stable estimates and plots.

R packages used:

CAUSE (*Morrison et al., 2020*)
TwoSampleMR (*Hemani et al., 2018*)
RadialMR (*Bowden et al., 2018*)
Meta R (*Schwarzer et al., 2015*)

Post-review analysis: during the peer-review process we performed further analysis using CAUSE method for body fat percentage (BFP), waist circumference adjusted for BMI, and waist-to-hip ratio adjusted for BMI.

