## [Editor Report]

This paper is of potential interest to those researchers and clinicians working in the area of physical activity and obesity. The authors have presented strong evidence of a causal relationship between physical activity and higher BMI. The conclusions that have been made are supported by the data and are translationally relevant.

---

## [Decision Letter]

**Decision letter after peer review:**

Thank you for submitting your article "Mendelian randomization suggests a bidirectional, causal relationship between physical inactivity and obesity" for consideration by *eLife*. Your article has been reviewed by 3 peer reviewers, and the evaluation has been overseen by Y M Dennis Lo as the Senior Editor/Reviewing Editor. The following individuals involved in review of your submission have agreed to reveal their identity: Xin He (Reviewer #2); Ashley Budu-Aggrey (Reviewer #3).

Essential revisions:

1) It is mentioned in the abstract and shown in Table 1 that BMI lead to increase in sedentary but not moderate/more vigorous activities. Could the authors explain why it may be the case (one may expect no associations for all or consistent associations for these 3)?

2) The use of BMI is reasonable but did the authors consider other measures of (central) obesity, eg waist circumference, waist-hip ratio, fat % etc., for which GWAS summary data should be available.

3) The no. of SNPs should be presented in table 2.

4) How do the MR estimate in this study compare to effect sizes of physical exercise (or interventions of exercise program) from previous clinical (epidemiology) studies? How does this study support or refute findings from previous works? Please kindly elaborate further.

5) Typo? in the conclusion part: increasing sedentary time is likely to be beneficial for weight management.

6) Supp Text p21 line 471

I don't think the inside assumption can be checked by I-squared (it addresses NOME assumption mainly). Please refer to

https://www.ncbi.nlm.nih.gov/pmc/articles/PMC5446088/.

7) It may be useful to visualize the MR analysis with scatter plot of the effect sizes of variants on the exposure (BMI) and outcome (sedentary time). In the plot, the variants can be colored by their contribution to the CAUSE statistics, see Figure 4 of the CAUSE paper. This plot would help show, for example, whether there are outlier variants; or whether the results are largely driven by just a small number of variants.

8) CAUSE is susceptible to false positives when the value of q, a measure of the proportion of shared variants, is high. The authors stated that q is about 0.2, which is pretty small. However, it is unclear if this is q under the causal model or the sharing model. If q is small under the sharing model, the result would be quite convincing. This needs to be clarified.

9) For transparency, it would be useful to make the genetic instruments, the summary data and the scripts used to perform the analysis publicly available. It would be useful to emphasize in the manuscript that it is the causal effect of genetic liability for these traits that have been investigated. In addition, the authors may want to mention that further study would be warranted to investigate causal relationships at various BMI thresholds.

*Reviewer #1 (Recommendations for the authors):*

The author employed a causal inference method known as Mendelian randomization (MR) to evaluate potential causal relationships between physical activities and obesity. The strengths and part of the limitations are given in the public review part.

Here are my further suggestions/comments for authors' consideration

1) It is mentioned in the abstract and shown in Table 1 that BMI lead to increase in sedentary but not moderate/more vigorous activities. Could the authors explain why it may be the case (one may expect no associations for all or consistent associations for these 3)?

2) The use of BMI is reasonable but did the authors consider other measures of (central) obesity, eg waist circumference, waist-hip ratio, fat % etc. , for which gwas summary data should be available

3) I suggest the no. of snps be presented in table 2

4) How do the MR estimate in this study compare to effect sizes of physical exercise (or interventions of exercise program) from previous clinical (epidemiology) studies? How does this study support or refute findings from previous works? Please kindly elaborate further.

5) typo? in the conclusion part: increasing sedentary time is likely to be beneficial for weight management.

6) Supp Text p21 line 471

I don't think the inside assumption can be checked by I-squared (it addresses NOME assumption mainly). Please refer to

https://www.ncbi.nlm.nih.gov/pmc/articles/PMC5446088/.

*Reviewer #2 (Recommendations for the authors):*

None.

*Reviewer #3 (Recommendations for the authors):*

The authors have presented a well written and conclusive study, where strong evidence suggests a causal bidirectional relationship between BMI and physical inactivity.

The methods used and results presented do support the conclusions that have been claimed in this study.

For transparency, I would suggest making the genetic instruments, the summary data and the scripts used to perform the analysis publicly available. I would also stress that it is the causal effect of genetic liability for these traits that have been investigated. In addition, the authors may want to mention that further study would be warranted to investigate causal relationships at various BMI thresholds.

---

## [Author Response]

Essential revisions:1) It is mentioned in the abstract and shown in Table 1 that BMI lead to increase in sedentary but not moderate/more vigorous activities. Could the authors explain why it may be the case (one may expect no associations for all or consistent associations for these 3)?

This is a very relevant point to clarify. Sedentary time reflects the time spent awake in sedentary behaviors involving a sitting, reclining or lying posture. Moderate physical activity reflects the time spent in brisk walking, slow bicycling or other behaviors that affect heart rate and breathing. Sedentary time and physical activity are thus not opposite of each other, and individuals can be physically active even if they would increase their daily time spent in sedentary behaviors. Therefore, it is important to address the causal relationship with obesity separately for sedentary time and physical activity.

We have now clarified this aspect in the Abstract and Introduction:

Lines 58-60: “It is important to assess the causal relationship with adiposity separately for physical activity and sedentary time, as individuals can be physically active even if they engaged a substantial part of their time to sedentary behaviors, and vice versa (Panahi and Tremblay, 2018).”

2) The use of BMI is reasonable but did the authors consider other measures of (central) obesity, eg waist circumference, waist-hip ratio, fat % etc., for which GWAS summary data should be available.

Thank you, this is a very good suggestion. We have now expanded our analyses using CAUSE to other measures of obesity and central obesity, including body fat percentage (BFP) and waist circumference and waist-hip ratio adjusted for BMI (WC_adjBMI_ and WHR_adjBMI_). Overall, we find that BFP shows consistent results with BMI, but find no evidence of causality between physical inactivity or sedentary time and WC_adjBMI_ or WHR_adjBMI_.

We have reported these new results in Lines 235-241:

“The results from CAUSE analyses for BFP were consistent with those for BMI, i.e. there was evidence of a causal effect of higher vigorous and moderate physical activity on lower BFP (P=1.4x10-6 and P=0.004, respectively), and a causal effect of more sedentary time on higher BFP (P=0.009) (Appendix 1-table 6 and 7, Figure 2, Appendix 1-figure 2). We found no evidence of a causal relationship between physical activity or sedentary time and measures of central adiposity, including WCadjBMI and WHRadjBMI (Appendix 1-table 6 and 7, Figure 2, Appendix 1-figure 2).”

We have described the methods used in these new analyses in Lines 373-379:

“In addition to BMI, we assessed causal effects of physical activity and sedentary time on body fat percentage (BFP) and on two measures of central adiposity, including waist circumference and waist-hip ratio adjusted for BMI (WCadjBMI and WHRadjBMI, respectively). GWAS results for BFP were available from an analysis of 454,633 participants of European ancestry in the UK Biobank (Elsworth et al., 2020). GWAS results for WCadjBMI and WHRadjBMI were available from GIANT Consortium meta-analyses of 231,355 and 694,649 individuals of European ancestry, respectively (Justice et al., 2017, Pulit et al., 2019).”

To note, since we are now examining four different continuous measures of adiposity in the manuscript, we have changed the word “obesity” to “adiposity” in the title and elsewhere in the manuscript, whenever needed and justified.

3) The no. of SNPs should be presented in table 2.

We have now added the number of SNPs in Table 2.

4) How do the MR estimate in this study compare to effect sizes of physical exercise (or interventions of exercise program) from previous clinical (epidemiology) studies? How does this study support or refute findings from previous works? Please kindly elaborate further.

We have now elaborated further on the effect sizes of physical activity and exercise interventions on weight loss in previous studies, and compared these findings with the causal estimates observed in the present Mendelian randomization analyses:

Lines 278-294: “Our results are well in accordance with a previous observational study that aimed to assess the bidirectional relationship between physical activity and weight change during a 10-year period (Barone Gibbs et al., 2020). Examining associations between accelerometer-based activity measures and weight change in 866 men and women, the study suggested a bidirectional relationship where higher sedentary time at baseline increased 10-year weight gain and higher baseline weight was associated with an unfavorable 10-year change in sedentary time. The effect sizes indicated that 1 hour lower sedentary time at baseline was associated with ~0.3 kg decrease in body weight over the 10-year follow up, whereas 3 kg higher baseline body weight was associated with ~2 min increase in sedentary time, closely resembling the causal estimates we observed in the present Mendelian randomization analyses. Our results are also in accordance with randomized clinical trials of physical activity interventions which generally suggest that increasing physical activity leads to a moderate loss of body weight in overweight or obese participants (Church et al., 2009, Rosenkilde et al., 2012, Golubic et al., 2015, Kim et al., 2019, Biddle et al., 2015). Resembling the causal estimates we observed, meta-analyses of randomized clinical trials have suggested weight losses ranging from 0.3 kg to 1.8 kg for various physical activity interventions ranging from 2-52 weeks (Twells et al., 2021).”

5) typo? in the conclusion part: increasing sedentary time is likely to be beneficial for weight management.

Thank you for noting this error. We have now corrected the sentence to “decreasing sedentary time is likely to be beneficial for weight management” (lines 347-348).

6) Supp Text p21 line 471I don't think the inside assumption can be checked by I-squared (it addresses NOME assumption mainly). Please refer tohttps://www.ncbi.nlm.nih.gov/pmc/articles/PMC5446088/.

We agree and have now clarified that we checked the NOME assumption, not the inside assumption, using I-squared and used the suggested reference (PMC5446088) in Lines 670-672:

“The no measurement error (NOME) assumption, which assumes no measurement error between the genetic instrument and the exposure, was checked by calculating the mean F-statistic and the variation of the I2 statistic [3].”

7) It may be useful to visualize the MR analysis with scatter plot of the effect sizes of variants on the exposure (BMI) and outcome (sedentary time). In the plot, the variants can be colored by their contribution to the CAUSE statistics, see Figure 4 of the CAUSE paper. This plot would help show, for example, whether there are outlier variants; or whether the results are largely driven by just a small number of variants.

We agree and have now added a scatter plot of the expected log pointwise posterior density (ELPD) contributions of each variant to BMI and sedentary time, and the contributions of the variants to selecting either the causal model or the shared model (Supplementary Figure 1 panel A). We identified one clear outlier variant (red circle) that we thus decided to remove before re-running the CAUSE analysis (panel B). We found that the causal effect of BMI on sedentary time remained of similar magnitude before and after the removal of this outlier variant (β=0.13, P=6x10^-4^ and β=0.13, P=3x10^-5^, respectively) (Appendix 1-table 1 and 2).

We have added a paragraph in the Results section to describe these new findings:

Lines 204-210: “We checked for outlier variants by producing a scatter plot of expected log pointwise posterior density (ELPD) contributions of the variants to BMI and sedentary time (Appendix 1-table 1), identifying one clear outlier variant (rs6567160 in MC4R gene) (Figure 2, Appendix 1-figure 1). However, the causal effect of BMI on sedentary time remained consistent even after removing this outlier variant from the CAUSE analysis (Appendix 1-table 1 and 2).”

8) CAUSE is susceptible to false positives when the value of q, a measure of the proportion of shared variants, is high. The authors stated that q is about 0.2, which is pretty small. However, it is unclear if this is q under the causal model or the sharing model. If q is small under the sharing model, the result would be quite convincing. This needs to be clarified.

We thank the reviewer for a very relevant question. We have now clarified in the manuscript that all of the reported q values (~0.2) were under the causal model (lines 202-203). We applied the strict parameters for the priors in CAUSE in all of our analyses, which leads to high shared model q values (q=0.7-0.9). To examine whether our bidirectional causal findings for BMI and sedentary time may represent false positive results, we performed a further analysis to identify and exclude outlier variants, as described in our response to Question 7. I.e. we produced a scatter plot of expected log pointwise posterior density (ELPD) contributions of each variant to BMI and sedentary time, and the contributions of the variants to selecting either the causal model or the shared model (Supplementary Figure 1 panel A). We identified one clear outlier variant (red circle) that we thus removed (panel B), but the magnitude of the causal estimates was not affected by the exclusion of the variant (Appendix 1-table 1 and 2).

9) For transparency, it would be useful to make the genetic instruments, the summary data and the scripts used to perform the analysis publicly available. It would be useful to emphasize in the manuscript that it is the causal effect of genetic liability for these traits that have been investigated. In addition, the authors may want to mention that further study would be warranted to investigate causal relationships at various BMI thresholds.

Thank you for the valuable comments. We fully agree that making the data and scripts publicly available is important. Thus, we have now added a data sharing statement in the end of the manuscript along with the web-link and github repositories where the summary data, genetic instruments and R scripts are freely available

Lines 461-466: “Data sharing: All analyses were performed using R statistical package freely available at https://cran.r-project.org/mirrors.html. The CAUSE R package and instructions are available at https://jean997.github.io/cause/. The Two-sample MR package is available at https://mrcieu.github.io/TwoSampleMR/. The RadialMR package is available at https://github.com/WSpiller/RadialMR. The code and curated data for the current analysis are available at https://github.com/MarioGuCBMR/MR_Physical_Activity_BMI.”

As suggested by the reviewers, we have now also clarified that we investigated the causal effect of genetic liability for physical activity and adiposity traits.

Lines 24-25: “We assessed genetic liability using results from genome-wide association studies for accelerometer-based physical activity and sedentary time…”

Lines 67-68: “Here, we aimed to assess the causality between the genetic liability of physical inactivity, sedentary time and BMI…”

Lines 266-267: “The present Mendelian randomization analyses suggest a bidirectional causal relationship between higher genetic liability of sedentary time and higher BMI…”

We have now also added a sentence in the Discussion section to state that future research is needed to investigate the casual relationships in connection to different BMI thresholds.

Lines 342-343: “Moreover, further research is needed to investigate causal relationships at various BMI thresholds.”

Reviewer #1 (Recommendations for the authors):The author employed a causal inference method known as Mendelian randomization (MR) to evaluate potential causal relationships between physical activities and obesity. The strengths and part of the limitations are given in the public review part.Here are my further suggestions/comments for authors' consideration1) It is mentioned in the abstract and shown in Table 1 that BMI lead to increase in sedentary but not moderate/more vigorous activities. Could the authors explain why it may be the case (one may expect no associations for all or consistent associations for these 3)?

This is a very relevant point to clarify. Sedentary time reflects the time spent awake in sedentary behaviors involving a sitting, reclining or lying posture. Moderate physical activity reflects the time spent in brisk walking, slow bicycling or other behaviors that affect heart rate and breathing. Sedentary time and physical activity are thus not opposite of each other, and individuals can be physically active even if they would increase their daily time spent in sedentary behaviors. Therefore, it is important to address the causal relationship with obesity separately for sedentary time and physical activity.

We have now clarified this aspect in the Abstract and Introduction:

Lines 58-60: “It is important to assess the causal relationship with adiposity separately for physical activity and sedentary time, as individuals can be physically active even if they engaged a substantial part of their time to sedentary behaviors, and vice versa (Panahi and Tremblay, 2018).”

2) The use of BMI is reasonable but did the authors consider other measures of (central) obesity, eg waist circumference, waist-hip ratio, fat % etc. , for which gwas summary data should be available

Thank you, this is a very good suggestion. We have now expanded our analyses using CAUSE to other measures of obesity and central obesity, including body fat percentage (BFP) and waist circumference and waist-hip ratio adjusted for BMI (WC_adjBMI_ and WHR_adjBMI_). Overall, we find that BFP shows consistent results with BMI, but find no evidence of causality between physical inactivity or sedentary time and WC_adjBMI_ or WHR_adjBMI_.

We have reported these new results in Lines 235-241:

“The results from CAUSE analyses for BFP were consistent with those for BMI, i.e. there was evidence of a causal effect of higher vigorous and moderate physical activity on lower BFP (P=1.4x10-6 and P=0.004, respectively), and a causal effect of more sedentary time on higher BFP (P=0.009) (Appendix 1-table 6 and 7, Figure 2, Appendix 1-figure 2). We found no evidence of a causal relationship between physical activity or sedentary time and measures of central adiposity, including WCadjBMI and WHRadjBMI (Appendix 1-table 6 and 7, Figure 2, Appendix 1-figure 2).”

We have described the methods used in these new analyses in Lines 373-379:

“In addition to BMI, we assessed causal effects of physical activity and sedentary time on body fat percentage (BFP) and on two measures of central adiposity, including waist circumference and waist-hip ratio adjusted for BMI (WCadjBMI and WHRadjBMI, respectively). GWAS results for BFP were available from an analysis of 454,633 participants of European ancestry in the UK Biobank (Elsworth et al., 2020). GWAS results for WCadjBMI and WHRadjBMI were available from GIANT Consortium meta-analyses of 231,355 and 694,649 individuals of European ancestry, respectively (Justice et al., 2017, Pulit et al., 2019).”

To note, since we are now examining four different continuous measures of adiposity in the manuscript, we have changed the word “obesity” to “adiposity” in the title and elsewhere in the manuscript, whenever needed and justified.

3) I suggest the no. of snps be presented in table 2

We have now added the number of SNPs in Table 2.

4) How do the MR estimate in this study compare to effect sizes of physical exercise (or interventions of exercise program) from previous clinical (epidemiology) studies? How does this study support or refute findings from previous works? Please kindly elaborate further.

We have now elaborated further on the effect sizes of physical activity and exercise interventions on weight loss in previous studies, and compared these findings with the causal estimates observed in the present Mendelian randomization analyses:

Lines 278-294: “Our results are well in accordance with a previous observational study that aimed to assess the bidirectional relationship between physical activity and weight change during a 10-year period (Barone Gibbs et al., 2020). Examining associations between accelerometer-based activity measures and weight change in 866 men and women, the study suggested a bidirectional relationship where higher sedentary time at baseline increased 10-year weight gain and higher baseline weight was associated with an unfavorable 10-year change in sedentary time. The effect sizes indicated that 1 hour lower sedentary time at baseline was associated with ~0.3 kg decrease in body weight over the 10-year follow up, whereas 3 kg higher baseline body weight was associated with ~2 min increase in sedentary time, closely resembling the causal estimates we observed in the present Mendelian randomization analyses. Our results are also in accordance with randomized clinical trials of physical activity interventions which generally suggest that increasing physical activity leads to a moderate loss of body weight in overweight or obese participants (Church et al., 2009, Rosenkilde et al., 2012, Golubic et al., 2015, Kim et al., 2019, Biddle et al., 2015). Resembling the causal estimates we observed, meta-analyses of randomized clinical trials have suggested weight losses ranging from 0.3 kg to 1.8 kg for various physical activity interventions ranging from 2-52 weeks (Twells et al., 2021).”

5) Typo? in the conclusion part: increasing sedentary time is likely to be beneficial for weight management.

Thank you for noting this error. We have now corrected the sentence to “decreasing sedentary time is likely to be beneficial for weight management” (lines 347-348).

6) Supp Text p21 line 471I don't think the inside assumption can be checked by I-squared (it addresses NOME assumption mainly). Please refer tohttps://www.ncbi.nlm.nih.gov/pmc/articles/PMC5446088/.

We agree and have now clarified that we checked the NOME assumption, not the inside assumption, using I-squared and used the suggested reference (PMC5446088) in Lines 670-672:

“The no measurement error (NOME) assumption, which assumes no measurement error between the genetic instrument and the exposure, was checked by calculating the mean F-statistic and the variation of the I2 statistic [3].”

Reviewer #3 (Recommendations for the authors):The authors have presented a well written and conclusive study, where strong evidence suggests a causal bidirectional relationship between BMI and physical inactivity.The methods used and results presented do support the conclusions that have been claimed in this study.For transparency, I would suggest making the genetic instruments, the summary data and the scripts used to perform the analysis publicly available. I would also stress that it is the causal effect of genetic liability for these traits that have been investigated. In addition, the authors may want to mention that further study would be warranted to investigate causal relationships at various BMI thresholds.

Thank you for the valuable comments. We fully agree that making the data and scripts publicly available is important. Thus, we have now added a data sharing statement in the end of the manuscript along with the web-link and github repositories where the summary data, genetic instruments and R scripts are freely available

Lines 461-466: “Data sharing: All analyses were performed using R statistical package freely available at https://cran.r-project.org/mirrors.html. The CAUSE R package and instructions are available at https://jean997.github.io/cause/. The Two-sample MR package is available at https://mrcieu.github.io/TwoSampleMR/. The RadialMR package is available at https://github.com/WSpiller/RadialMR. The code and curated data for the current analysis are available at https://github.com/MarioGuCBMR/MR_Physical_Activity_BMI.”

As suggested by the reviewers, we have now also clarified that we investigated the causal effect of genetic liability for physical activity and adiposity traits.

Lines 24-25: “We assessed genetic liability using results from genome-wide association studies for accelerometer-based physical activity and sedentary time…”

Lines 67-68: “Here, we aimed to assess the causality between the genetic liability of physical inactivity, sedentary time and BMI…”

Lines 266-267: “The present Mendelian randomization analyses suggest a bidirectional causal relationship between higher genetic liability of sedentary time and higher BMI…”

We have now also added a sentence in the Discussion section to state that future research is needed to investigate the casual relationships in connection to different BMI thresholds.

Lines 342-343: “Moreover, further research is needed to investigate causal relationships at various BMI thresholds.”